# Meta-Learning and Meta-Reinforcement Learning - Tracing the Path towards DeepMind's Adaptive Agent

## Abstract

Humans are highly effective at utilizing prior knowledge to adapt to novel tasks, a capability that standard machine learning models struggle to replicate due to their reliance on task-specific training. Meta-learning overcomes this limitation by allowing models to acquire transferable knowledge from various tasks, enabling rapid adaptation to new challenges with minimal data. This survey provides a rigorous, task-based formalization of meta-learning and meta-reinforcement learning and uses that paradigm to chronicle the landmark algorithms that paved the way for DeepMind's Adaptive Agent, consolidating the essential concepts needed to understand the Adaptive Agent and other generalist approaches.

## 1 Introduction

Humans excel at reusing prior knowledge to adapt rapidly to novel tasks. In contrast, standard machine learning models are typically trained for a single task on a large amount of task-specific data. Optimized for task-specific peak performance, they excel in trained domains but struggle to generalize to new tasks. Meta-learning (or learning how to learn) seeks to overcome this issue by extracting higher-level knowledge and strategies from a distribution of distinct yet related tasks, enabling models to adapt quickly to new challenges with minimal additional data.

Historically, the conceptual groundwork for self-adapting artificial intelligence was laid by (Schmidhuber, 1987). Numerous studies in the subsequent decades established the theoretical foundations of meta-learning (Sutton, 2022), e.g., by incorporating inductive bias shifts into the framework (Schmidhuber et al., 1997) (see Vanschoren (2018), Huisman et al. (2021), Sutton (2022) for a detailed historic review). The field experienced another resurgence, concurrent with the rise of deep learning, when substantially larger datasets and computational resources became available. By introducing Model-Agnostic Meta-Learning (MAML), (Finn et al., 2017a) proposed the first landmark algorithm in the class of gradient-based meta-learners, the abstraction of gradient-based learning to the meta-level (see Section 3.1). Around the same time, (Duan et al., 2016) introduced $RL^2$, the first landmark of the class of memory-based learners (see Section 3.2). Since then, meta-learning techniques have diversified across few-shot image classification (He et al., 2023), (Gharoun et al., 2024), (Li et al., 2021), neural architecture search (Elsken et al., 2019), (Hospedales et al., 2022), (Ren et al., 2021), (Pereira, 2024), natural language processing (Yin, 2020), (Lee et al., 2022), (Lee et al., 2021), reinforcement learning (see Section 2.2), and application fields like

- robotics, where meta-learners learn novel strategies from only a few demonstrations (Finn et al., 2017b), or experiences (Johannsmeier et al., 2019).

- healthcare (Rafiei et al., 2024), where patient- or disease-specific data is often sparse (Tan et al., 2022), (Maicas et al., 2018).

- adaptive control (McClement et al., 2022), (Duanyai et al., 2024), where system parameterizations change over time.

Even for the preparation of space missions (Gaudet et al., 2020), researchers use meta-learning to pre-train models that can rapidly adapt in real time to changing environmental conditions. Consequently, several works

survey meta-learning by categorizing different sub-classes of meta-learning (Beck et al., 2023b), (Vettoruzzo et al., 2024), differentiating between the numerous related topics (Vettoruzzo et al., 2024), (Barcina-Blanco et al., 2024), (Upadhyay, 2023), addressing open problems (Beck et al., 2023b), or reviewing overlaps between meta-learning and its most related fields (Upadhyay, 2023).

However, the literature lacks a compact, rigorous mathematical treatment tying meta-learning theory to practical implementations: Performance measures are frequently illustrated informally (e.g., in figures or captions) but are rarely defined mathematically, which complicates fair comparisons between methods. Additionally, practical concepts such as validation or meta-validation are seldom formalized. This issue is particularly acute in meta-RL, where agents' actions influence data collection; accordingly, cumulative reward must be explicitly incorporated into the meta-objective. But a careful understanding of meta-learning and meta-RL paradigms is highly relevant - perhaps more than ever. As large black-box foundation models and generalist agents scale, they increasingly exhibit emergent capabilities that reproduce behaviours previously engineered via bespoke training schemes (see Section 4). Without precise formalisms and metrics, it remains difficult to assess whether and to what extent such capabilities constitute genuine meta-learning - a problem that is particularly acute for newcomers to the field.

**Contribution and Paper Outline**

The primary goal of this work is to close the gap in formalism by providing a rigorous formalization of meta-learning and meta-RL within the task-based paradigm. This survey thus offers newcomers an accessible entry point into the field. Meta-learning is distinguished from its closest neighbors in Appendix B; however, related areas such as continual learning, self-supervised learning, and active learning remain outside the scope. For these topics, readers are referred to (Vettoruzzo et al., 2024) or to dedicated surveys. Furthermore, this work does not address metric-based meta-learning, as it contributes little to meta-RL and the Adaptive Agent. Those specifically interested in this sub-field are directed to other surveys such as (Vanschoren, 2018) or (Huisman et al., 2021). By contrast, the central contribution of this survey is a chronological presentation of the landmark developments culminating in generalist agents such as DeepMind's Adaptive Agent (ADA), using the task-based meta-learning paradigm as a unifying framework to consolidate the essential knowledge. For this purpose, this work is organized as follows:

- It derives meta-learning from standard supervised learning (Section 2.1) and systematically transfers the resulting notions and formulas to the meta-RL setting (Section 2.2). As part of this unified mathematical framework, Section 2.3 carefully defines the most important meta-learning performance measures that are treated informally in the literature. Additionally, the mathematical derivation of Bayesian Reinforcement Learning - which is not a central concept of this work - can be found in Appendix A.

- In Section 3, this work applies the formal paradigm as a consistent interpretive lens to present the landmarks on the path from early meta-learners to ADA. The paradigm of Section 2 serves as the organizing thread: each new algorithm is situated within the same formalism to clarify how the field evolved. The resulting timeline begins with the simpler family of gradient-based meta-learners (Section 3.1), then presents memory-based algorithms. The memory-based sequence starts with the simpler class of RNN-based meta-learners (Section 3.2) and then gradually increases in complexity until the Adaptive Agent (Section 3.5), is finally presented.

- It analyzes ADA as a representative meta-RL instance of a large-scale generalist agent; thereby discussing the key scaling and enhancement techniques (e.g., distillation, automated curriculum learning), and positions these mechanisms within the formal paradigm to explain how they contribute to ADA's capabilities.

Finally, this work examines emergent capabilities, interpretability challenges, and open problems in Section 4, before giving an outlook framed by the "Three Pillars of General Intelligence", and concluding in Section 5.

## 2 Paradigm

This section formally introduces meta-learning and meta-RL by comparing them to standard machine and reinforcement learning respectively. It, thereby, creates a consistent structure of terms, notions and performance measures (in Section 2.3 used throughout the rest of the entire work.

### 2.1 Meta-Learning

In standard machine learning a learner $f_\theta$ with parameters $\theta$ is trained to solve a particular task $T$ of the form [1]

$$T := (\mathcal{L}, \mathcal{X}, \mu, \mathbb{T}, h), \tag{1}$$

by minimizing the loss function $\mathcal{L} : \mathcal{X} \to \mathbb{R}$ on some training data $X_{\text{train}}$ out of the observation space $\mathcal{X} \subseteq \mathbb{R}^d$. The transition $\mathbb{T} : \mathcal{X} \times \mathcal{X} \to \mathbb{R}$ represents the transition from one observation to another [2], while $\mu$ denotes the initial distribution and $h$ the horizon of the task.

**Example 1 - Image Classification:**

Let $X$ be a labeled image set out of the observation space $\mathcal{X}$ of images showing cats or dogs, and let $X_{\text{train}}, X_{\text{val}}$ and $X_{\text{test}}$ be disjoint subsets of $X$ with a split of 80% training data and 10% validation and test data. Then, the task $T$ to classify images between 0 (dog) or 1 (cat) is formally defined by defining the components in (1), i.e.,:

- One can select any loss function $\mathcal{L}$ that is suitable for classification, e.g., the cross entropy loss.

- Since all images in $X_{\text{train}}$ are equally likely to be sampled, the initial distribution $\mu$ is the uniform distribution conditioned on $X_{\text{train}}$.

- The transition can be defined as the probability of sampling image pair $(x_1, x_2) \in \mathcal{X} \times \mathcal{X}$. However, the concrete definition depends on the specific type of sampling (with or without replacement). item Since each shot in image classification consists of only one image, the horizon $h$ equals 1 [a].

The goal is to find the optimal function $f^*$ correctly classifying any image as cat or dog [b] by minimizing $\mathcal{L}_\theta$ w.r.t. $\theta$. The notation $\mathcal{L}_\theta$ denotes that observations $x \in \mathcal{X}$ are processed or collected via the function $f_\theta$. The learner $f_\theta$ can be any suitable machine learning model, e.g., a convolutional neural network processing images. This work focuses on deep learning so that the parameters $\theta$ are assumed to always correspond to the weights of a deep neural network throughout. The validation and test sets $X_{\text{val}}$ and $X_{\text{test}}$ contain labeled cat and dog images excluded from training. They allow one to evaluate whether the learner $f_\theta$ truly learned to distinguish cats from dogs or only overfitted by memorizing the images in $X_{\text{train}}$. Validation runs during training to monitor progress; testing runs once after training to report final performance. But this way of testing the model $f_\theta$ does not provide any information if the same learner could also distinguish between cats and horses since the latter class does not exist in the task's observation space $\mathcal{X}$.

---

[a]In sampling-based tasks, such as classification, horizon, transition, and initial distribution play a rather theoretical role. Therefore, they are only shown here for the sake of completeness. Example 2 illustrates the case of a sequence-based task, where those components are more meaningful.

[b]The optimal function $f_T^*$ is not necessarily unique, but can, without loss of generality, be assumed as one particular function, as each optimal function, is per definition, as good as any other.

The loss function $\mathcal{L}$ as well as all other task-specific components, such as observation space $\mathcal{X}$ or transition $\mathbb{T}$, remain unchanged throughout the whole training and testing process. As a consequence, the model $f_\theta$ is

---

[1]This formalization of a task is inspired by (Finn et al., 2017a), (Upadhyay, 2023), (Rakelly et al., 2019).

[2]Note that this is the most general way of defining the transition function. In many cases, it is defined as the probability distribution of observing observation pairs $(x_1, x_2) \in \mathcal{X} \times \mathcal{X}$. Hence, one must, technically, define the transition as $\mathbb{T} : \sigma(\mathcal{X} \times \mathcal{X}) \to [0, 1]$. However, as all observations $x \in \mathcal{X}$ should generally be measurable, the $\sigma$ function is redundant notation

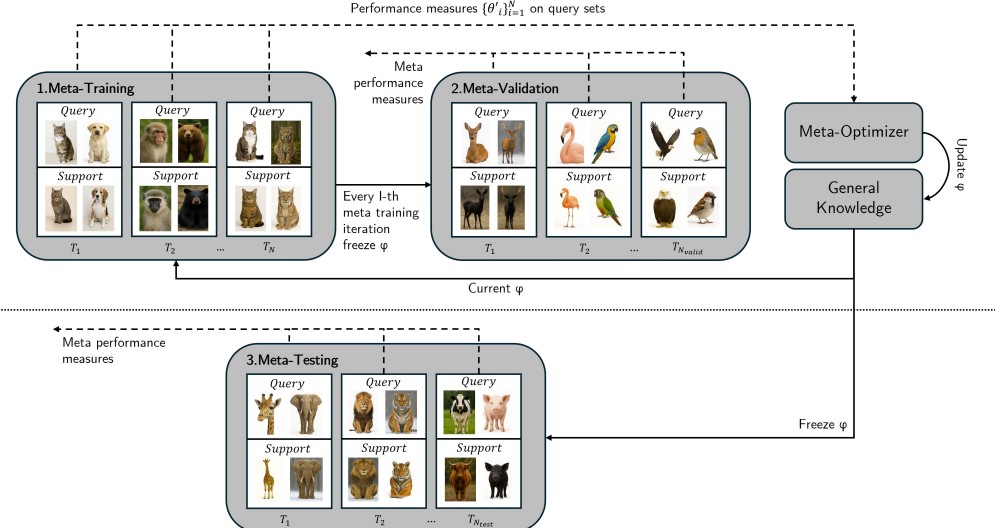

Figure 1: Meta-learning of 2-way 1-shot animal classification tasks. The current meta-knowledge $\varphi$ is the prior for one-shot learning of each particular classification task. During meta-training, the meta-optimizer receives all $N$ query set losses of the adapted models to update meta-knowledge $\varphi$. Meta-validation evaluates the training progress on new classification problems every $l$ meta-epochs, while meta-testing on unseen classifications takes place after meta-training.

tailored to solve the task $T$, while it generally fails to generalize to out-of-domain tasks, i.e., to tasks with different loss, transition, or observation space (Upadhyay, 2023). However, for many tasks the number of training examples available is not sufficient for learning, and sometimes training a (standard) learner for a new but very similar task fully from scratch is too costly in terms of computation power or training time.

For these reasons, meta-learning - unlike standard machine learning - seeks to capture general knowledge across a family of similar tasks, enabling rapid, task-specific adaptation, i.e., adaptation to a task within $K$ shots rather than with a full training [3]. Modifying the observation space $\mathcal{X}$ of Example 1 to contain labeled images of several different animals, but only ten images per class, one obtains a meta-learning problem. With the same training-validation-test split as in Example 1, this means eight images per class for training and only one for validation and testing. This small amount of training data does not suffice for training a standard learner properly. However, on a meta-level, distinguishing between all these different animals does not differ much from the task in Example 1. Quite the opposite is the case since, on an abstract level, classifying dogs vs. cats requires similar high-level skills as classifying cats vs. horses: The model needs to identify the creature in the picture first, recognize shapes of noses, ears, or other body parts, examine the silhouette, etc. In other words, classifying cats vs. horses is just another task from the task distribution over different animal classification tasks of the form presented in Example 1. In this way, the meta-task to classify any animal can be separated into several simplified tasks that share some common, high-level structure and belong to the same "meta-problem" of classifying animals. The following paragraphs formalize the meta-learning paradigm, while Figure 1 applies it to the meta-problem of classifying different animals.

**The Meta-Learning Paradigm**

Formally, the meta-learning paradigm (Thrun & Pratt, 1998) consists of a distribution $p(T)$ over tasks of the form

$$T_i := (\mathcal{L}_i, \mathcal{X}, \mu_i, \mathbb{T}_i, h). \tag{2}$$

Within this framework, each task of the form (2) may possess its own loss function, transition dynamics, or initial distribution, while the observation space $\mathcal{X}$ and the horizon $h$ are generally assumed to be identical

---

[3]If only one shot or even no shot is used for fine-tuning, one speaks of one-shot or zero-shot learning, respectively.

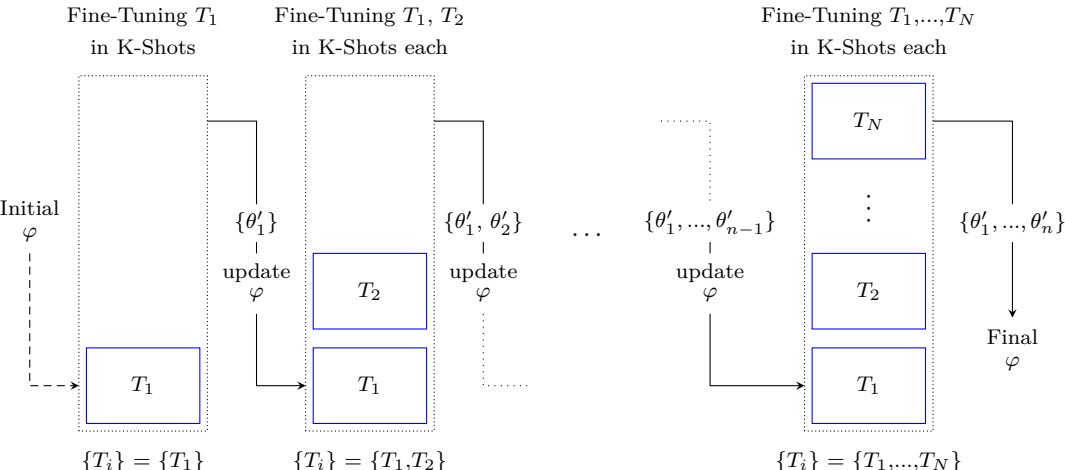

Figure 2: General Meta-Training. In each iteration a new task $T_i$ is sampled from the family $p(T)$. The meta-variable $\varphi$ is the prior for individual $K$ shot fine-tuning of each task. The resulting parameters $\theta_i'$ of each task are used to update $\varphi$.

across tasks drawn from the same distribution $p$ (see e.g., Finn et al. (2017a), Rakelly et al. (2019)). This work follows this convention, although it is not a necessary condition. However, some works specifically focus on domain generalization (Li et al., 2018), (Triantafillou et al., 2020). The definition of the observation space highly depends, among other factors, on the underlying problem, and so does the meaning of the notion "'identical'': In the meta-task of classifying animals, it can be defined to contain all images of animals that can potentially occur within the distribution $p(T)$ of different animal classification tasks. But the format (e.g., grey-scale vs. RGB images) needs to be identical. For other task distributions, e.g., distributions of RL tasks as discussed in Section 2.2, defining the observation space might mean something completely different.

In the task-based paradigm, there are two components of what must be learned from a theoretical point of view: Common knowledge over all tasks $T_i$ in $p(T)$, and task-specific knowledge gained through few-shot fine-tuning. Hence, the formal learning paradigm consists of two stages, the more abstract meta-level and the few-shot standard learning of a particular task $T_i \sim p(T)$. This paradigm is explicitly applied to all gradient-based meta-learning algorithms presented in Section 3.1. However, even for the memory-based meta-learners presented in Sections 3.2 and 3.4 that do not explicitly divide learning into meta-learning and task-specific standard learning, the meta-training paradigm can be formalized as (implicitly) two-stage by introducing a meta-variable $\varphi$ encoding common knowledge over all tasks. As a consequence, the training-validation-test split is also two-stage. On meta-task level, certain tasks get explicitly excluded from the meta-training task pool to function as validation throughout meta-training or as test tasks after meta-training. The corresponding evaluation takes place on the stage of task-specific standard learning, which is why it is often referred to as inner learning. However, since every task $T_i$ from the distribution $p(T)$ is assumed to be a few-shot learning task, a task-specific validation set $X_{\text{val}}^i$ is not required. This way, standard few-shot learning is mimicked within each task $T_i$ while meta-training provides every such task-specific fine-tuning with a meaningful prior to enable fast adaptation.

**Meta-Training and Meta-Testing**

The corresponding meta-training scheme aims to optimize $\varphi$ to be the best prior for fast inner learning. As highlighted in Figure 2, this means yielding task-specific parameters from the general knowledge $\varphi$ and measure their test set performance after $K$ shots of task-level adaptation. The corresponding optimization problem in each meta-training iteration is

$$\text{Minimize} \quad \mathbb{E}_{T_i \sim p(T)} \mathcal{L}_{\text{meta}} \left( \theta_{T_i}^*(\varphi), \ \varphi, \ X_{\text{test}}^i \right) \quad \text{wrt. } \varphi. \tag{3}$$

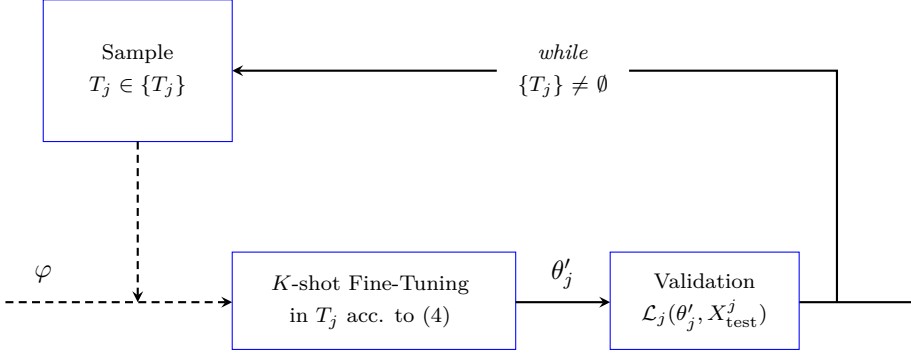

Figure 3: General Meta-Testing Paradigm. For each $T_j$ sampled from the set of test tasks the parameters $\theta_j(\varphi)$ are fine-tuned in $K$ shots, before the resulting $\theta'_j$ get evaluated on the task's test set via the task-specific loss $\mathcal{L}_j$ to yield the performance.

The notation $\theta_i(\varphi)$ highlights the relationship between $\varphi$ and the initial parameters of the inner learning, while $\mathcal{L}_{\text{meta}}$ denotes the meta loss function and $\theta_i^*(\cdot)$ the optimal task-specific parameters for Task $T_i$ when starting inner learning with meta-parameter $\varphi$. However, the optimal task-specific parameters $\theta_i^*(\cdot)$ are rather a theoretical component of the formalization of meta-training taken from (Upadhyay, 2023) than a practical implementation.

Inner learning aims to approximate the task-specific optimal parameters $\theta_i^*$ representing the optimal solution of the task $T_i$. Thus, the resulting parameters must be denoted differently to distinguish them from the optimal ones. Throughout this work, the respective notation is $\theta'_i := \theta'_i(\varphi)$ for parameters trained in inner learning starting from $\theta_i(\varphi)$ i.e., from initial parameters $\theta$ yielded from the prior $\varphi$. The resulting optimization problem for inner learning is

$$\text{Minimize} \quad \mathcal{L}_i\left(\theta_i, \theta_i(\varphi), X_{\text{train}}^i\right) \quad \text{wrt. } \theta_i. \tag{4}$$

The choice of the meta-loss $\mathcal{L}_{\text{meta}}$ and the inner learning determines the respective meta-learning framework. This holds for all algorithms presented in Section 3, although the meta-loss is not explicitly given for the memory-based meta-learners in Section 3.2. Moreover, the task sampling throughout meta-training is algorithm-specific. Figure 2 shows a scheme where tasks are iteratively drawn from $p(T)$ up to $N$ tasks, but other algorithms might directly start with $N$ tasks and re-sample them, respectively. The numbers $N$, $N_{\text{val}}$ and $N_{\text{test}}$ of training, validation, and testing tasks are themselves hyperparameters of meta-training.

After meta-training a meta-model $f_\varphi$, meta-testing aims to evaluate the trained model on unseen test tasks. However, evaluating the quality of a meta-model means examining how well it boosts task-specific learning. Hence, the task-specific parameters $\theta_j(\varphi)$ are adapted within $K$ shots of inner learning for each test task, while only the meta-variable $\varphi$ is fixed throughout the whole meta-testing process. Evaluating the correspondingly adapted task-level parameters $\theta'_j$ on the task-level test set $X_{\text{test}}^j$ yields the task-level performance required for the meta-performance measures presented in the next section. Figure 3 shows this iterative meta-testing scheme.

## 2.2 Meta-Reinforcement Learning

This subsection transfers the general meta-learning paradigm to that of meta-Reinforcement Learning (meta-RL). Mimicking the structure of Section 2.1, it starts with presenting the standard reinforcement learning (RL) paradigm and thereafter derives the meta-RL paradigm from it.

### Standard Reinforcement Learning

Standard RL is a special case of the standard machine learning paradigm, where an agent interacts with an environment $(\mathbb{A}, \mathbb{S}, R)$ by selecting an action $a \in \mathbb{A}$ based on the state $s \in \mathbb{S}$ and receiving the next state $s' \in \mathbb{S}$ and a reward signal $r \in \mathbb{R}$. Example 2 illustrates this agent-environment interaction, where the

transition of a general task (1) extends to a distribution $\mathbb{T} : \mathbb{S} \times \mathbb{A} \times \mathbb{S} \to \mathbb{R}$ to additionally denote the effect of the agent's action on the subsequent state. The reward function $R : \mathbb{A} \times \mathbb{S} \to \mathbb{R}$ and the discount factor $\gamma \in [0, 1]$ determine the task loss $\mathcal{L}$. The reward as well as the transition function must fulfill the Markov property so that the standard task (1) forms a Markov Decision Process (MDP)

$$M := (\mathbb{A}, \mathbb{S}, R, \gamma, \mathbb{T}, \mu, h), \tag{5}$$

with action space $\mathbb{A}$, state space $\mathbb{S}$, episode horizon $h$ [4] and initial state distribution $\mu$. However, as the underlying MDP $M_i$ determines the corresponding RL task $T_i$, these terms will be used synonymously throughout this work.

**Example 2 - Racing games:**

Considering a racing game, where the agent's goal is to drive in a way that finishes a racing track as fast as possible, the different components of the underlying MDP (5) are defined as follows:

- The set of possible actions $\mathbb{A}$ contains all directions in which the racer can move, its speed, and whether or not to brake.

- The state space $\mathbb{S}$ contains all states the racer can possibly find itself in. Additional to the current racing track (as pixels), it may include information on the racer's speed as well as the time since the start of the race or other useful information a player might have.

- The initial distribution over states $\mu$ refers to what the racer "sees" in its starting position. If that position is deterministic, $\mu$ is deterministic, too.

- The reward function can easily be modeled as one when reaching the goal and zero otherwise.

- The discount factor $\gamma$ must be smaller than one to encourage the agent to finish the racing track as fast as possible.

- The transition models the effect any action $a \in \mathbb{A}$ has on any state $s$ [a]. Since most racing tracks are entirely deterministic, this connection is also likely to be deterministic.

- Each race is an episode. Hence, the time the agent needs to finish the racing track determines the horizon $h$, which is therefore dynamic.

The strategy $\pi$ is the theoretical decision rule of the agent. It is representative of the agent, as it determines, how the racer actually drives at any time in the race.

---

[a]Technically, the general transition introduced in the previous section is the probability of observing tupel $(s, a, s') \in \mathbb{S} \times \mathbb{A} \times \mathbb{S}$. However, in most cases this simplified version suffices.

The loss $\mathcal{L}$ of a RL task is often defined as the value function [5]

$$\mathcal{L} := -\mathbb{E}_{\pi_\theta(\cdot)}^{s_0 \sim \mu} \sum_{t=0}^{h} \gamma^t r_{t+1} \tag{6}$$

where $r_{t+1} = R(s_t, a_t)$ and $\mathbb{E}_{\pi_\theta(\cdot)}^{s_0 \sim \mu}$ denotes the expectation under the assumptions that $s_0$ is drawn from the initial distribution $\mu$ and every action $a_t$ is taken according to policy $\pi_\theta(s_t)$. Correspondingly, in a RL task, the goal is to find an action selection strategy $\pi : \mathbb{S} \to \mathbb{A}$ that maximizes the reward collected throughout an episode of agent-environment interaction.

As this work only considers deep RL, the policy $\pi_\theta$ is defined by the agent's neural network weights $\theta$. There are almost as many rules for updating $\theta$ as there are RL algorithms, and not all of them explicitly minimize

---

[4]RL problems do not have to be episodic, but $h < \infty$ is a rather practical condition.
[5]The agent does not necessarily have to only update its parameters at the end of an episode, although this is assumed throughout this paper. However, generalizing the value function to the expected future reward starting from any state $s_t$ at any time $t$ only requires for an index shift within the sum.

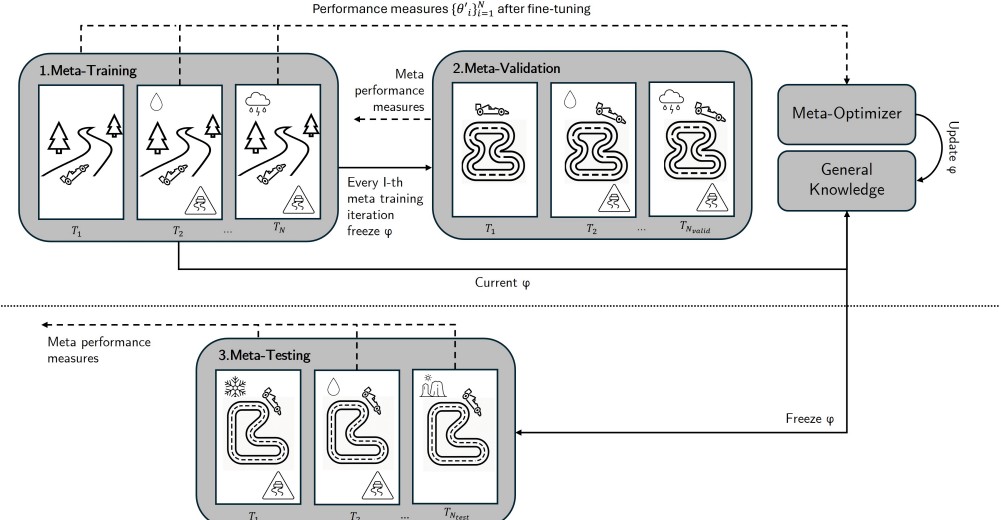

Figure 4: Meta-Reinforcement learning to race on tracks with varying weather conditions. Starting from the meta-knowledge $\varphi$, $K$ episodes of fine-tuning on the particular racing track yield performance measures. The meta-optimizer uses these measures to update meta-knowledge $\varphi$. Meta-validation evaluates the training progress on unseen racing tracks every $l$ meta-epochs. After meta-training, the meta-policy adapts to test tracks to evaluate the quality of prior $\varphi$.

(6) (Sutton & Barto, 2018), (Francois-Lavet et al., 2018). For example, the family of function approximation RL algorithms aims to approximate the loss (6) (or modifications of it) by optimizing $\theta$ to best approximate the loss. Afterwards, they achieve the loss minimization by acting accordingly.

To learn the optimal strategy $\pi_i^*$ for a particular MDP $M_i$, an agent must explore the environment to uncover new actions and their potential rewards. At the same time, it must exploit its gained knowledge to maximize immediate rewards. This trade-off is called the exploration-exploitation dilemma, and it motivates the Bayesian RL paradigm presented in Appendix A.

**The Meta-Reinforcement Learning Paradigm**

Interacting with an environment can be extremely financially expensive e.g., in robotics, trading or transportation. For such problems, an agent must be trained within a simulation and, thereafter, adapt as fast as possible to the real world. Moreover, environmental dynamics often change over time e.g., when shocks permanently increase market prices, when a robot is assigned to a different task within the same physical environment or when a racing track becomes more slippery due to heavy rain. In such scenarios, a standard RL agent must often be trained fully from scratch, although it has already gained a lot of knowledge about the environment that is still valid. For these reasons, meta-RL aims to collect general knowledge over a family of similar MDPs.

One can easily modify Example 2 into such a meta-RL problem by parameterizing the racing track's slipperiness. For each different value of slipperiness the transition $\mathbb{T}$ of the underlying MDP slightly changes, while the overall structure of state and action spaces and even reward remain unchanged. Such families of MDPs are called parametric. Acting optimally in any of these environments requires almost the same high-level skills: The agent still needs to learn driving dynamics, when to brake and which direction to drive to, just with a different level of drifting in curves due to a change in slipperiness. In other words, the meta-problem to finish a randomly slippery racing track as fast as possible (see Figure 4 for an illustration), remains the same.

Formally, the meta-RL paradigm - analogous to general meta-learning - consists of a distribution $p(M)$ over MDPs of the form

$$M_i := (\mathbb{S}, \mathbb{A}, R_i, \mathbb{T}_i, \mu_i, \gamma, h). \tag{7}$$

The MDPs from the same family $\{M_i\}_{M_i \sim p(M)}$ normally vary with respect to their underlying dynamics $\mathbb{T}_i$ or reward function $R_i$, while sharing some structure like state and action space - which, together, can be interpreted as the observation space $\mathcal{X}$ of a general task (2). However, these assumptions are more practical than a theoretical necessity.

Collecting knowledge within a family of MDPs (possibly) means two similar but different things: Collecting general knowledge about structures like state or action space shared within the family (see sections 3.2 and 3.4), or identifying the current MDP as fast as possible to adjust the agent's policy $\pi_\theta$ accordingly (see sections C and 3.3). For racing MDPs of the form presented in Example 2 the former means learning how to process the pixels shown as the current state, remembering general information such as the route of the race, and understanding how to drive a car on an abstract level. The latter corresponds to the exploration/exploitation dilemma on MDP-level. It requires determining as quickly as possible how slippery the racing track actually is in order to adjust the style of driving accordingly. As a consequence, meta-RL is therefore a two-stage process: On meta-level common knowledge over all MDPs is collected, while on MDP-level the current MDP is identified to adjust the MDP-specific policy. The training-validation-test split is analogous to that of meta-learning.

**Meta-Training and Meta-Testing**

Both the inner optimization (4) as well as the outer optimization (3) are straightforward to adjust to the meta-RL training scheme by introducing a meta-variable $\varphi$ representing meta-knowledge. The only core structural difference is in the MDP-specific training and testing data $X_{\text{train}}^i$ and $X_{\text{test}}^i$: As the agent has to interact with its environment to observe the subsequent state and reward, sampling an observation set $X \in \mathcal{X}$ always depends on the current policy. one typically refers to this as "collecting experience" of the form $X = \{s_t, a_t, r_{t+1}, s_{t+1}\}_{t=0}^h$. Consequently, in the meta-RL paradigm, gathering data from a particular MDP $M_i$, in $K$ shots, means collecting $K$ episodes of experience within the environment following the strategy $\pi_{\theta_i(\varphi)}$ that might or might not change between episodes depending on the inner RL algorithm.

The meta-testing scheme in meta-RL is also analogous to the general meta-testing scheme. The only key difference is in the MDP-level testing of the adapted policy $\pi'_{\theta_j}$ as it interacts with its environment for another $K_{\text{test}}$ episodes after inner learning without further adaptation. The accumulated (and normalized) reward gained throughout these test episodes represents the test loss $\mathcal{L}_{\text{test}}^j$ required for calculating the performance measures presented in the previous section.

## 2.3 Performance Measures

In contrast to standard learning, where performance is typically evaluated by measuring a learner's loss on unseen test data, assessing a meta-learner's performance is more challenging (see the Appendix of Shala et al. (2024b) for a full portfolio of respective plots and tables). Task-level test performance alone is often insufficient to fully characterize adaptation capability during meta-learning. Instead, various measures must be used to draw a more detailed picture of the adaptation capability the model $f_\varphi$ developed throughout meta-training. However, in many works (e.g., Rakelly et al. (2019), Norman & Clune (2024), Duan et al. (2016)), the notion "performance" exclusively refers to the total loss collected within the task-level test batches of test tasks. Similarly, the notion "asymptotic performance" only refers to the task-level test loss after a full, standard training [6] (see e.g., Rakelly et al. (2019), where it is only implicitly defined). Other measures such as adaptation speed or sample-efficiency are only implicitly shown in tables or graphs (see, e.g., Zintgraf et al. (2020), Melo (2022), Shala et al. (2024b)). This makes it difficult to immediately understand results, particularly for newcomers, and impossible to quantify performance across different works. To address these issues, the following paragraphs define, motivate, and discuss the most important meta-learning performance measures.

---

[6] The notion "asymptotic" implies $K \to \infty$, which, from a theoretical point of view, is what happens in standard learning.

**Generalization:** In standard learning, generalization refers to the ability of a learner to perform well on unseen data after training. It is generally quantified by the generalization error, i.e., the difference between the learner's performance on the training set and its performance on the test set:

$$\mathcal{L}_{\text{gen}}(\theta) := \mathcal{L}(\theta, X_{\text{test}}) - \mathcal{L}(\theta, X_{\text{train}}). \tag{8}$$

Analogously, the generalization ability of a meta-learner $f_\varphi$ can be quantified by the accumulated generalization error over the unseen meta-test tasks:

$$\mathcal{L}_{\text{gen}}^{\text{acc}}(\varphi) := \mathcal{L}_{\text{test}}^{\text{meta}}(\varphi) - \mathcal{L}_{\text{train}}^{\text{meta}}(\varphi), \tag{9}$$

where $\theta'_K(\varphi)$ denotes the adapted parameters after $k$ training shots on the respective task $T_j$ starting from prior $\varphi$. The accumulated train and test losses are defined as the sum over the respective (standard learning) train and test losses

$$\mathcal{L}_{\text{train}}^{\text{meta}}(\cdot) := \sum_{T_j \sim p(T)} \mathcal{L}(\cdot, X_{\text{train}}^j), \qquad \mathcal{L}_{\text{test}}^{\text{meta}}(\cdot) := \sum_{T_j \sim p(T)} \mathcal{L}(\cdot, X_{\text{test}}^j),$$

which must be normalized to avoid differently scaled losses throughout different tasks.

In RL, however, comparing results is even more difficult, since observed rewards depend on the current state, the time, the actions taken by the agent, and how the reward function is modelled. Normalizing rewards (e.g. through min-max-scaling) does not equalize the distribution of potential rewards throughout different reward functions, which are, potentially, designed to encode completely different objectives (potentialy implying different optimal strategies). Probably, this is why all meta-RL algorithms presented along the timeline in section 3 examine reward curves on different tasks individually. However, it is still possible to gain a basic understanding of a learner's generalization ability as long as tasks from the same distribution have similar reward functions that represent similar objectives. The generalization performance is simply not quantifiable.

The accumulated generalization error (9) measures the generalization performance at task level. In contrast, one measures meta-generalization by abstracting (9) to the meta-generalization error

$$\mathcal{L}_{\text{gen}}^{\text{meta}}(\varphi) := \mathcal{L}_{\text{gen}}^{\text{test}}(\varphi) - \mathcal{L}_{\text{gen}}^{\text{train}}(\varphi), \tag{10}$$

where $\mathcal{L}_{\text{gen}}^{\text{train}}$ and $\mathcal{L}_{\text{gen}}^{\text{test}}$ denote the accumulated generalization error (9) summed over all meta-training tasks or meta-test tasks, respectively. The meta-generalization error evaluates how well the model $f_\varphi$ generalizes on the meta-level by taking its task-specific generalization capability on the training tasks into account. A good meta-generalization means that $\varphi$ boosts task-specific learning equally well for training and test tasks, while a bad meta-generalization hints $\varphi$ to overfit to the training tasks in such a way, that fine-tuning has the maximal success, but this knowledge cannot be transferred to other tasks from the same distribution. The latter is called meta-overfitting.

**Adaptation speed:** Adaptation speed refers to the rate at which a learner can effectively learn new tasks $T_j$ from a limited number of training steps. It is typically measured by tracking task-specific metrics over test task training iterations and examining the slope of the gained curve (see e.g., Finn et al. (2017a), Figures 2 and 5, Melo (2022), Figure 5, Shala et al. (2024b), Figure 4). A high slope indicates fast adaptation, and vice versa. In the context of meta-learning, one typically tracks adaptation performance during the individual training of each test task $T_j$, i.e., the task-specific training loss $\mathcal{L}_{\text{train}}^j$ [7]. This way, one yields an adaptation function $\mu_{\varphi, T_j}^{\text{adapt}} : K \to \mathbb{R}$ with

$$\mu_{\varphi, T_j}^{\text{adapt}}(k) := \mathcal{L}_{\text{train}}^j \left( \theta_j^{(k)}, X_{\text{train}}^{j,(k)} \right), \tag{11}$$

whose derivative w.r.t. the number $k$ of inner learning epochs (e.g., gradient descent steps) is the adaptation speed on the task $T_j$. Calculating (11) for each test task $T_j$ and each fine-tuning iteration $k$ yields the meta-adaptation performance. However, since the adaptation function (11) is generally not known, one must

---

[7]Note that, in theory, any task-specific loss can be used for evaluating the adaptation speed. For example, using the task-specific generalization (8) one examines, how fast a model gains generalization capabilities within one particular task $T_j$. However, this is rather complicated and, hence, none of the works presented in this survey utilize such a sophisticated version of adaptation speed.

approximate it by empirically calculating it for different values of $k$ (see, e.g., Finn et al. (2017a), Tables 1 and 2). In RL, this typically means evaluating (11) episodewise (see, e.g., Zintgraf et al. (2020), Figure 5).

**Sample-Efficiency:** Since gathering data is often difficult or expensive, it is also desirable to develop meta-learners that adapt to unseen situations without the need for large amounts of data. Such learners are referred to as sample-efficient. In contrast to adaptation speed, which measures how fast a model can learn and improve within a certain amount of training iterations regardless of the samples needed, sample-efficiency is about learning effectively from fewer examples regardless of the training iterations required for extracting the data's information. It can be measured by [8]

$$\mu_{\varphi, T_j}^{\text{sample eff}}(S) := \mathcal{L}_{\text{train}}^j \left( \theta_j^{(S)}, X_{X \subseteq X_{\text{train}}^j}^{|X|=S} \right), \tag{12}$$

where $\theta_j^{(S)}$ denotes the parameters $\theta_j(\varphi)$ fine-tuned with $S$ observations. Similar to adaptation, one is typically more interested in the rate at which performance increases for additional samples, i.e., in the (empirical) derivative of (12). In classification, for example, the sample efficiency is directly determined by varying the number $K$ of shots (see e.g., Finn et al. (2017a), Raghu et al. (2020), Tables 1 and 5), while, in RL, it typically corresponds to the amount of experience, i.e., number of timesteps, collected by the task-specific agent.

**Out-of-distribution:** While generalization refers to a learner's ability to perform well on unseen data drawn from the same distribution as the training data, out-of-distribution (OOD) performance specifically evaluates how well a model can handle data that comes from a different distribution. In other words, while generalization measures how well the learner can apply learned patterns to new examples within the same context, OOD assesses a learner's ability to generalize to situations not represented during training. Consequently, high OOD performance indicates robustness and adaptability, while poor performance can lead to failures in real-world applications where conditions may vary significantly from the training scenarios. On meta-level, this means defining the test tasks as the family $\{T_j\}_{T_j \not\sim p(T)}$ of tasks, that are not drawn from the task distribution $p(T)$. Then, measuring the performance means calculating all the metrics mentioned above on these OOD tasks.

## 3 The Timeline of meta-Reinforcement Learning Landmarks

This section traces the evolution of meta-RL algorithms up to DeepMind's Adaptive Agent, starting with the class of gradient-based meta-learners and then moving to memory-based approaches. Each subsection presents the landmark algorithm of the respective class on the timeline towards ADA, describing the related literature, meta-training and meta-testing schemes, and performance analyses for each landmark algorithm in clearly distinguished paragraphs. Following the previously established paradigm as a guiding framework, each landmark introduces a new concept, allowing the overall complexity to grow gradually towards ADA. Except for gradient-based methods (Section 3.1), the rest of this section focusses entirely on meta-RL. For gradient-based meta-learning, an additional paragraph hence derives the corresponding meta-RL variant from the more general meta-learning approach. The broader family of memory-based (black-box) meta-RL algorithms splits into two subsections: RNN-based methods (Section 3.2) and transformer-based methods (Section 3.4). In between, Section 3.3 discusses task inference, as its landmark (VariBAD) is an RNN-based algorithm extended by Bayesian inference to model task uncertainty. The section closes with ADA, the current state-of-the-art meta-RL algorithm, a generalist, transformer-based agent that integrates common techniques for boosting meta-training such as distillation and auto-curriculum learning.

Table 1 summarizes the advantages and drawbacks of the components presented along the landmarks in this section.

### 3.1 Gradient-based Meta-Learning

Deep neural networks are gradient-based models, i.e., optimized with stochastic gradient descent (SGD) or modifications like Adam (Kingma & Ba, 2017) or AdamW (Loshchilov & Hutter, 2018). But these optimizers

---

[8]Note that, similar to adaptation speed, sample-efficiency can be measured with any kind of task-specific loss.

often converge slowly or even fail to converge. Convergence depends heavily on the starting point: starting near the optimum can yield rapid progress, while random initialization often requires many more iterations or leads to a poor local minimum. A second issue is the choice of the learning rate. A rate that is too high can cause instability or oscillation, while choosing it too low can significantly slow down learning (Nar & Sastry, 2018).

Both these problems could be tackled, if it were possible to a priori place the parameters $\theta$ in a sufficiently narrow area around the optimal solution $\theta^*$. Not only would learning be more stable and directed towards that optimum, but a small learning rate would also suffice to approach it within a few steps. This is the main idea of gradient-based meta-learning [9].

### Model-Agnostic Meta-Learning

Model-Agnostic Meta-Learning (MAML) (Finn et al., 2017a) is the foundational gradient-based meta-learning algorithm and one of the earliest and simplest landmarks in the evolution of meta-learning and meta-RL. The main idea is to place the meta-level prior $\varphi$ in a region that is promising for all tasks drawn from the distribution $p(T)$, so that starting each task-specific fine-tuning from that prior, i.e., setting $\theta_i(\varphi) = \varphi$ for all $\theta_i$, $K$ gradient descent steps maximize task-specific performance during fine-tuning. It is broadly applicable to any machine-learning problem where the loss function is sufficiently smooth to compute gradients. As a consequence, many works have successfully adapted the MAML paradigm to various machine learning domains such as neural architecture search (Wang et al., 2022a), regression (Sen & Chakraborty, 2024), multi-object tracking (Chen & Deng, 2024), time-series classification (Wang et al., 2024a), and semi-supervised learning (Boney & Ilin, 2018), while applying it to various fields like medicine (Tian, 2024), (Alsaleh et al., 2024), (Tian et al., 2024), (Ranaweera & Pathirana, 2024), (Naren et al., 2021), biomass energy production (Zhang et al., 2025), or fault diagnoses in bearings (Lin et al., 2023).

### Meta-Training and Meta-Testing

As the meta-training of MAML directly derives from the general meta-training scheme presented in Section 2.1, it consists of a meta-level and a task-specific stage. The meta-loss $\mathcal{L}_{\text{meta}}$ is defined as the sum over all task-specific losses $\mathcal{L}_i$ on the respective test sets $X_{\text{test}}^i$:

$$\mathcal{L}_{\text{meta}} := \sum_{i=1}^{N} \mathcal{L}_i \left( \theta_i', \ \varphi, \ X_{\text{test}}^i \right)$$

with $\theta_i'$ denoting the task-specific parameters resulting from the individual inner update steps (14). For $K = 1$, the corresponding meta-level gradient descent update is

$$\varphi' = \varphi - \beta \nabla_{\varphi} \mathcal{L}_{\text{meta}} \tag{13}$$

with pre-defined meta-learning rate $\beta$. This way, the meta-level gradient descent update of $\varphi$ implicitly optimizes $\varphi$ w.r.t. the performance of the inner learning (4), that, itself, is solved by $K$ gradient descent steps of the form

$$\theta_i' = \varphi - \alpha \nabla_{\varphi} \mathcal{L}_i \left( \varphi, X_{\text{train}}^i \right), \tag{14}$$

with a pre-defined learning rate $\alpha$[10], and the meta-level parameters $\varphi$ as the corresponding task-level prior. Figure 5 shows one iteration of the MAML meta-training scheme with $K = 1$. However, as the meta-testing of MAML fully follows the scheme presented in Figure 3, no additional figure shows the MAML meta-testing scheme.

To avoid the computation of second-order gradients, (Finn et al., 2017a) suggest First-Order (FO) MAML, a more efficient approach that omits second-order derivatives. Although this modification does not guarantee

---

[9]Although initialization schemes, such as the Glorot (Glorot et al., 2011), He (He et al., 2015), or LeCun (LeCun et al., 1989) initializations, already stabilize initial training, they, in contrast to gradient-based meta-learning, do not actively place model parameters in an area which is promising for all tasks of a task distribution $p(T)$.

[10](Finn et al., 2017a) state that the learning rate $\alpha$ can also be meta-learned, which is addressed in $\alpha$-MAML (Behl et al., 2019)

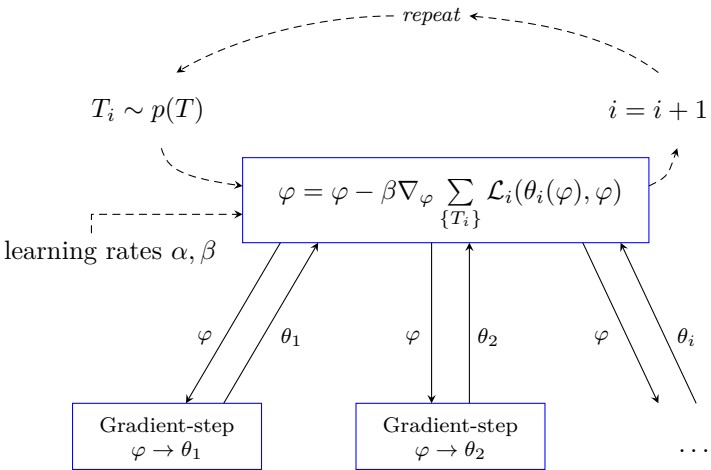

Figure 5: The MAML meta-training scheme.

global convergence (Fallah et al., 2020) (a limitation that motivated further modifications like Hessianfree MAML (Fallah et al., 2020) or Implicit MAML (Rajeswaran et al., 2019)), FO-MAML is much easier to implement and computationally less expensive than memory-based meta-learning algorithms presented in the subsequent sections (Finn et al., 2017a).

### The MAML Meta-RL Scheme

Besides the fact that task-specific training and test sets must be represented by $K$ episodes of MDP-specific agent-environment interaction, the general MAML structure of embedding task-specific gradients in the meta-level gradient descent remains the same in meta-RL. However, the gradient descent steps (14) and (13) require gradients of the task-specific losses, which are the task-specific expected returns (6) in meta-RL. But these value functions are not differentiable due to unknown environment dynamics and the agent's impact on them, so that the original MAML meta-RL paradigm can only apply policy-gradient methods (Sutton et al., 1999). This makes MAML on-policy. While the exact computation of the task-level policy gradient depends on the respective task-level algorithm, the meta-update of $\varphi$ requires policy gradients $\nabla_\varphi \pi_{\theta_i'}$ of the updated parameters $\theta_i'$ on the respective task-specific test episodes $X_{\text{test}}^i$. In this way, the underlying policy-gradient algorithm is extended to the meta-level, i.e. to updating $\varphi$ with respect to the performance of the fine-tuned parameters $\theta_i'$ on test sets $X_{\text{test}}^i$. But this meta-update scheme consists of second-order policy gradients which are difficult to derive, what motivates meta-RL-specific first-order MAML extensions such as Taming-MAML (Liu et al., 2019) or DICE (Foerster et al., 2018).

Many state-of-the-art standard RL algorithms, such as PPO (Schulman et al., 2017) or TRPO (Schulman, 2015), utilize an off-policy learning scheme to learn from many policies at once. This motivates the PEARL (Rakelly et al., 2019) algorithm, the landmark gradient-based off-policy meta-RL algorithm used as a baseline for various sophisticated meta-RL approaches. It utilizes a replay buffer for offline meta-level updates while task-level policy gradients are not required. Instead, inner learning is informed via embeddings of the currently collected experience. However, PEARL does not directly contribute to the development path towards the Adaptive Agent, so that the curious reader is referred to Appendix C for a full derivation of PEARL's paradigm and meta-training scheme.

### Performance Analysis

Assuming that MAML's model architecture consists of a sufficiently deep neural network with ReLU activations and that the loss function is either cross-entropy or mean squared error (MSE), (Finn & Levine, 2018) prove that MAML serves as a universal function approximator for any training set $X_{\text{train}}^i$ and test set $X_{\text{test}}^i$ within an arbitrary task $T_i$. They show, furthermore, that MAML converges linearly to a global optimum

under MSE loss when using an over-parameterized deep neural network along with a SGD optimizer like Adam (Kingma & Ba, 2017). This implies that a deeper model, with at least one hidden layer, is necessary, even if single tasks can be addressed using a shallower or linear model (Arnold et al., 2021). Empirical findings support that MAML's meta-training can achieve 100% training accuracy (indicating global convergence) on simple task distributions ( such as Omniglot) when appropriate hyperparameters are utilized (Finn et al., 2017a), (Wang et al., 2022a). However, (Raghu et al., 2020) show that MAML learns effective feature representations rather than a rapidly adaptable prior: During task-specific training, the initial layers of the underlying network exhibit minimal changes, suggesting that the fundamental feature representations remain stable. Probably, this is why, MAML is more stable than other landmarks, especialy VariBAD (see Section 3.3), which is illustrated, e.g., on Meta-World Yu et al. (2020) Tasks (see Melo (2022), Figure 4), where MAML achieves almost 100% success rate, despite high task-uncertainty.

The number of different tasks to be learned simultaneously in the MAML paradigm is naturally bounded by the fact that model size cannot be increased indefinitely. Moreover, the MAML paradigm assumes the optimal parameters $\theta_1^*, \theta_2^*, \ldots$ of the different tasks $T_1, T_2, \ldots$ to be sufficiently close to each other, so that a general prior $\varphi$ (i.e., a global optimum) can be found from which each optimal solution can be sufficiently well approximated within $K$ gradient descent steps. But as parameter vectors are likely to be high-dimensional, this is a rather strong assumption. In meta-RL, the time MAML requires for learning RL tasks, scales with their complexity (Parisotto et al., 2019) since more complex tasks require better exploration and exploitation. Empirically, this is demonstrated on Mujoco (Todorov et al., 2012) tasks, such as HalfCheetahVel (see Melo (2022), Figures 4 and 5), where MAML achieves an average reward of $-150$ (averaged over different seeds), while memory-based algorithms, such as VariBAD (see Section 3.3, TrMRL (see Section 3.4, and RL$^2$ (see Section 3.2) achieve around $-25$ average reward starting from first episode. This, however, suggests a worse performance capability rather than a lower adaptation speed, since the rate of learning at the beginning is not significantly lower than in the other algorithms. This gap of performance is even more significant on OOD-tasks, such as illustrated in (Melo, 2022), Figure 6, where MAML achieves only $-500$ reward, while memory-based landmarks achieve a reward of approximately $-100$. These results provide the main motivation for the development of MAML extensions such as Robust MAML (Nguyen et al., 2021) and XB-MAML (Lee & Yoon, 2024), which are specifically designed to improve generalization to tasks that are slightly out of distribution."

## 3.2 Memory-based Meta-RL

In meta-RL, all MDPs are drawn from a meta-distribution $p(M)$ over MDPs that are distinct but similar. Memorizing similarities facilitates the adaptation to a new MDP that requires a similar strategy. In this respect, the identification of the current task $T_i$, i.e., the hidden dynamics of the corresponding MDP $M_i$, plays a crucial role in learning the task-specific optimal behaviour $\pi_i^*$. However, during MDP-level learning, this identification can only rely on the experience

$$c_t^i := \{(s_j, a_j, r_{j+1}, s_{j+1})\}_{j=1}^t \tag{15}$$

gathered until the current time $t$ in a particular MDP $M_i$. Such sequences are called context and numerous meta-learning algorithms such as PEARL (Rakelly et al., 2019), DREAM (Liu et al., 2021), (Gupta et al., 2018), MetaCure (Zhang et al., 2021), or CAVIA (Zintgraf et al., 2019), incorporate it into their paradigm.

How well the environment was explored is determined by the quality of the context (Norman & Clune, 2024), while a good representation of that context is substantial for good exploitation.

In Example 2, the current context is all segments of the racing track the racer has visited so far. Here, the ability to process the whole context sequence representing the racing track, rather than just single transitions, significantly helps to cover the dependencies between different sections of the race and to assign the reward - potentially occuring only once at the end of the race - to the whole route. Such problems, where time dependencies are important and rewards are sparse, motivate the use of memory architectures, as they are context-based sequence learners just by design.

The simplest memory architecture is that of a Recurrent Neural Network (RNN). RNNs maintain a hidden state $h$ that acts like a memory mechanism by storing information about past experience. They are Turing-

complete universal computers (Siegelmann & Sontag, 1992). However, even in standard RL, RNN-based agents are very sensitive to architectural choices like their initialization, the choice of the underlying RL algorithm, or the length of the context that is captured by the hidden state $h$ (Ni et al., 2022). The latter is, moreover, highly MDP-specific (Ni et al., 2022), which limits the flexibility of RNN-based meta-RL algorithms and makes them struggle to generalize between tasks (Beck et al., 2023a), (Ben-Iwhiwhu et al., 2022). This motivates additional modifications, such as employing a hypernetwork (Beck et al., 2023a) to learn initializations of the MDP-specific RNN weights, or incorporating Bayesian inference into the RNN-based meta-learners (as presented in Section 3.3). Nevertheless, the following paragraphs describe $RL^2$, the earliest and simplest memory-based meta-RL algorithm, that serves as the main reference for black-box (i.e., memory-based) meta-RL in many surveys, e.g., (Beck et al., 2023b), (Vettoruzzo et al., 2024), and as a baseline for most gradient-based and memory-based meta-learners.

**Fast Reinforcement Learning via Slow Reinforcement Learning**

The main idea of $RL^2$ is to combine a standard RL algorithm with an RNN-based agent [11]. The corresponding RNN weights update only once every $K$ episodes of MDP-specific learning, which aligns with the notion of "slow learning", while the activation of the RNN's hidden states by the current context serves as an implicit fast adaptation scheme throughout these $K$ episodes. This way, $RL^2$ can still be (theoretically) differentiated into an inner and an outer learning stage (Duan et al., 2016), although it consists of only one RNN representing the policy. The general knowledge $\varphi$ is implicitly collected by updating the weights of the RNN once every $K$ episodes. The goal to Bayes-optimally trade-off exploration and exploitation is also implicit: The objective of maximizing rewards over $K$ episodes results in the implicit need to explore the current MDP as fast as possible in order to maximize rewards in the subsequent episodes.

**Meta-Training and Meta-Testing**

The $RL^2$ meta-training scheme differs significantly from the general meta-RL meta-training scheme: One meta-episode is defined as interacting with a particular MDP $M_i$ for $K$ episodes. The experiences $X_{\text{train}}^i$ update the policy parameters $\theta$ in the standard RL manner on meta-level. For standard RL algorithms such as TRPO (Schulman, 2015) or PPO (Schulman et al., 2017), this means to sample single transitions $(s, a, r, s')$ from the replay buffer $\beta_i$ containing all experience in $X_{\text{train}}^i$, and use the corresponding batch for updating $\theta$ in several epochs of gradient descent.

Throughout MDP-specific learning, the parameters $\theta$ remain unchanged. Instead, the hidden state $h_t$ of the RNN updates through the gathered experience. It resets at the beginning of each MDP-specific training but gets preserved throughout the $K$ inner learning episodes - which is the major difference from simply modifying standard RL algorithms with an RNN agent. Conditioning the policy $\pi_\theta(\cdot|h_t)$ on that hidden state, one yields an MDP-specific context-based adaptation scheme, since the hidden state $h_t$ functions as the representation $z$ of the context $c$. An MDP-specific training-test split does not exist, as the inner learning performance is validated throughout MDP-specific training.

The meta-testing scheme of $RL^2$ is analogous to the meta-RL meta-testing presented in Section 2.2. The policy parameters $\theta$ are fixed, while the hidden state adapts during $K$ episodes of MDP-specific learning.

**Performance Analysis**

$RL^2$ was experimentally shown to be Bayes-optimal on simple benchmark tasks like $N$-arm bandit problems (Mikulik et al., 2020). The experiments indicated the Bayes-optimal solution to be a fixed point of meta-training. However, the task distributions used in (Mikulik et al., 2020) were manually constructed so that calculating a Bayes-optimal solution was tractable. Hence, the performance on broader, more complex task distributions is not guaranteed. In fact, $RL^2$ was empirically shown to achieve very poor asymptotic performance on locomotion tasks ((Gupta et al., 2018), Figure 4), Mujoco benchmark tasks ((Zintgraf et al., 2020), Figure 6, (Melo, 2022), Figure 4 and 5), and the MetaWorld environment ((Melo, 2022), Figure 5). On

---

[11]The originally published algorithm uses TRPO (Schulman, 2015) as the standard RL algorithm, although (Rakelly et al., 2019) show $RL^2$ to perform even better with more sophisticated RL algorithms like Proximal Policy Optimization (PPO) (Schulman et al., 2017).

all these tasks, the performance of $RL^2$ slightly increases at the beginning, but then it hardly learns anything else. This indicates poor generalization, as stated by (Ben-Iwhiwhu et al., 2022), probably, since the RNN has to cover all architectures theoretically needed for the optimal strategy $\pi_i^*$ of each MDP $M_i$ of the MDP distribution $p(M)$. However, $RL^2$ exhibits much better OOD performance (around $-150$ reward almost from the beginning) than MAML (not better than $-400$ reward) on Mujoco tasks (such as HalfCheetahVel) (Melo, 2022), Figure 6. Most likely, this suprisingly good OOD performance results from the high sample-efficiency and adaptation speed, that (Zintgraf et al., 2020) demonstrated for $RL^2$ on some Mujoco tasks (see Figures 5 and 6), but, reviewing the Figures presented in the literature, it is impossible to ultimately determine what the reason is.

### 3.3 Task-Inference Meta-RL

During learning, there is uncertainty about which MDP the agent is currently acting in. The actions taken by the current agent determine which experiences it gathers, but unseen states might be more informative about the current MDP and hence other actions might have been better for exploration. At the same time, the exploration-exploitation dilemma leads to poor reward during exploration, which is nevertheless required for exploiting the gained knowledge in order to maximize future rewards. As a consequence, many works, such as (Stadie et al., 2018), (Norman & Clune, 2024), (Liu et al., 2021), or (Zhang et al., 2021), separate exploration from exploitation by using distinct exploration and exploitation networks. Instead, the uncertainty about the underlying MDP can also be incorporated into the meta-RL framework by utilizing the Bayesian Reinforcement Learning (BRL) paradigm introduced in Appendix A. Thereby, the context updates the belief over the current MDP $M_i$ a posteriori, while the corresponding posterior distribution informs the decisions of the agent. Algorithms following this principle are referred to as task-inference methods (see, e.g., Beck et al. (2023b)). They appear both in the family of gradient-based meta-learners (Ni et al., 2022), (Ben-Iwhiwhu et al., 2022), (Gupta et al., 2018), (Zhang et al., 2021), and in the broader family of memory-based approaches (Zintgraf et al., 2020), (Bing et al., 2024). This subsection discusses Variational Bayes-Adaptive Deep RL (VariBAD) (Zintgraf et al., 2020), the landmark task-inference method on the development path towards the Adaptive Agent.

**Variational Bayes-Adaptive Deep RL**

The main idea of VariBAD is to extend the memory-based meta-RL framework presented in Section 3.2 to include task uncertainty by incorporating a Variational AutoEncoder (VAE). The corresponding VAE architecture consists of a context-encoder RNN $f_\varphi$ that outputs a variational distribution $p_\varphi(z|c_t^i)$, and an ordinary neural network decoder $g_\psi$ with two heads: One representing a common transition function $\mathbb{T}$, the other a common reward $R$. The policy $\pi_\theta(\cdot|s, p_\varphi(z|c))$ is, in contrast to $RL^2$, an ordinary deep neural network parameterized by $\theta$. Both the policy and the decoder receive the output distribution $p_\varphi(z|c)$ from the context-encoder $f_\varphi$, so that, in practice, both these components are different heads of the same architecture receiving the encoder output distribution $p_\varphi$ as input.

**Meta-Training and Meta-Testing**

Following the context-based meta-training scheme presented in Section 3.2, the meta-parameters $\varphi$, $\theta$, and $\psi$ do not adjust throughout the $K$ episodes of MDP-level training, while inner learning exclusively corresponds to the implicit change in the VAE distribution $p_\varphi(\cdot|c_t^i)$ resulting from updated context $c_T^i$. This way, VariBAD combines $RL^2$ with the PEARL algorithm (which is presented in Appendix C), since the VAE distribution directly incorporates task uncertainty into the model-based task-level decision-making. Additionally, the replay buffer $\beta$ stores the resulting trajectories $\{c_t^i\}_{t=1,\ldots,h_i}^{i=1,\ldots,N}$ for meta-level gradient descent updates [12]. The corresponding meta-loss $\mathcal{L}_{\text{meta}}$ consists of two major components:

1. The RL loss which simply is the accumulated loss over all trajectories, and

---

[12]Although the original publication (Zintgraf et al., 2020) utilizes on-policy algorithms, (Dorfman et al., 2021) present an off-policy modification, that is based on leveraging offline data collected by MDP-specific agents in order to extract meta-knowledge for meta-updates of $\varphi$, $\theta$, and $\psi$.

2. The ELBO loss, which itself consists of two components, a reconstruction loss for the decoder and a KL-divergence keeping the encoder output $p_\varphi$ close to its update[13].

Since the encoder output distribution $p_\varphi$ functions as the meta-learned prior, task inference takes place at the output stage of the VAE encoder. The posterior is updated at each time step via variational inference. However, the corresponding scheme is rather technical and hence not shown here. Instead, the curious reader is referred to the original work in (Zintgraf et al., 2020) or other works about task-inference like (Sajid et al., 2021). The meta-testing of VariBAD is analogous to the context-based meta-testing (Section 3.2).

**Performance Analysis**

VariBAD outperforms $RL^2$ in both a dynamic grid world and the MuJoCo benchmark (see Zintgraf et al. (2020), Figure 5). During the first five million time steps of the dynamic GridWorld environment, it achieves around twice as much reward, which indicates a two times higher sample-efficiency than $RL^2$ on that benchmark. Thereby, VariBAD exhibits almost Bayes-optimal exploratory behaviour (maximally 14% worse performance on the worst performance task in Figure 4). On Mujoco test tasks, it adapts within one single episode (see Zintgraf et al. (2020), Figure 5). Although $RL^2$ exhibits slightly better sample-efficiency at the beginning of fine-tuning on some of the Mujoco test tasks, VariBAD always adapts within one single episode, while other algorithms, such as PEARL (see Appendix C) cannot even solve some of the test tasks within the first two episodes. On other test tasks, VariBAD achieves three times as much reward as PEARL, i.e., exhibits a three times better adaptation speed, although PEARL shows significantly better sample-efficiency (see 3.3, Figure 6). This implies that VariBAD "takes time" to explore in early episodes, i.e. observes more frames to yield better episode reward at the cost of lower sample-efficiency. However, the adaptation speed of $RL^2$ on these tasks is similar to that of VariBAD (see Zintgraf et al. (2020), Figure 5b), which indicates a big role of the underlying RNN architecture and aligns with the performance analysis in Section 3.2.

The Bayes-optimality of VariBAD has neither been demonstrated nor proven across more sophisticated task distributions. In the MetaWorld environment, it exhibits unstable behavior throughout test task fine-tuning, i.e., the rate of successful performance oscilates between 0.5 and 0.8 (see Melo (2022), Figure 4), although it thereby outperforms TrMRL, the transformer-based landmark presented in Section 3.4. However, in OOD Mujoco tasks, VariBAD's instability leads to a much worse performance ( between $-100$ and $-300$ total reward) than that of TrMRL (constantly around $-100$ reward), see Melo (2022), Figure 6. According to Melo (2022), this indicates worse generalization than implied by the original publication Zintgraf et al. (2020). These issues led to modifications of VariBAD, such as HyperX (Zintgraf et al., 2021), which enhances meta-exploration by distilling policies from random networks to improve performance in distributions of sparsely rewarded MDPs, and MELTS (Bing et al., 2024), specifically designed for effective zero-shot performance on non-parametric task distributions through task-clustering. Both these methods utilize modifications that are also used in the Adaptive Agent (i.e., distillation and task-selection), which are hence described in Section 3.5.

### 3.4 Transformer-based Meta-RL

The Transformer network architecture was first proposed by (Vaswani et al., 2017) as a language-to-language model. As induced by its title, (Vaswani et al., 2017) show that "attention is all you need" by outperforming all other state-of-the-art language-to-language algorithms by a good margin. This success motivated the development of many Transformer modifications further boosting performance, e.g., the Transformer-XL (Dai et al., 2019) used in the Adaptive Agent. Simultaneously, several researchers developed modifications of the initial Transformer for tasks different from language-to-language translation, e.g., Vision Transformers for computer vision (Han et al., 2023), visual and semantic segmentation (Li et al., 2024a), (Xiao et al., 2023), (Strudel et al., 2021), forecasting (Lim et al., 2021), and RL (Hu et al., 2024). This way, they applied Transformers to various fields like medicine (Xiao et al., 2023), e.g., for RNA prediction (Chaturvedi et al., 2025), fault detection, e.g., in manufacturing (Wu et al., 2023), bioinformatics (Zhang et al., 2023a), automated driving (Hu et al., 2022), and many other application fields. In RL, Transformers close the

---

[13]For a detailed derivation of the ELBO-loss and all corresponding components, the curious reader is referred to (Sajid et al., 2021).

gap between simulation and reality in locomotion control (Lai et al., 2023), act beneficially in various IoT environments like Smart-Homes or industrial buildings (Rjoub et al., 2024), and learn from a vast amount of (sub)-optimal demonstrations (Liu & Abbeel, 2023), (Lee et al., 2023), (Reed et al., 2022). Since each of the various extensions and modifications of the vanilla transformer builds upon the attention mechanism proposed in the original publication (Vaswani et al., 2017), the original Transformer architecture is broadly presented and discussed in Appendix D, while this section focuses on the utilization of Transformer architectures for meta-RL.

**Transformers are Meta-Learners**

Instead of translating a sentence from one language into another, a context-based RL agent "translates" context trajectories of the form (18) into the current action $a_t$. Such a "translation" does not require a decoder as the current action can directly be derived from a policy head. But it is likely that the single transitions of a context trajectory have strong dependencies to each other or the time they occurred, so that it is additionally desirable to contextualize them. This motivates using the Transformer architecture, e.g., in simple, gradient-based meta-learning algorithms like MAML (Section 3.1). Hence, different works present MAML modifications for detecting faults in bearings (Li et al., 2024b), improving short-term load forecasting in scenarios where different clients require federated learning (Feng et al., 2025), and forecasting stock prices (Chen et al., 2025). However, Transformers additionally have a demonstrable long-term memory of up to 1500 steps into the past (Ni et al., 2023), which particularly motivates to use transformers in memory-based meta-learning (Melo, 2022), (Grigsby et al., 2023), (Xu et al., 2024), (Shala et al., 2024b), (Shala et al., 2024a).

TrMRL (Melo, 2022) is such a Transformer architecture tailored for meta-RL. It extends $RL^2$ by a Transformer architecture i.e., by self-attention, and fulfills all necessary properties of a meta-learner (Melo, 2022), i.e.,

1. The fast adaptation of the multi-head attention serves as a task representation mechanism, since each self-attention head contextualizes the embeddings of the context $c_t^i$ collected in the current MDP $M_i$ up until the current time step $t$.

2. the meta-learned model weights function as a long-term memory by design, through which the respective MDP can be identified and acted upon accordingly.

They hypothesize that any MDP can be represented by a distribution over working memories:

$$z_t = \sum_{t=1}^{T} \alpha_t f_\theta(o_t) \tag{16}$$

where $o_t$ is a single transition of the form $(s_t, a_t, r_{t+1}, s_{t+1})$, $T$ is the trajectory length, $f_\theta$ is a learnable, arbitrary linear function, and $\alpha_t$ are coefficients summing up to 1. In fact, the context scores of the last transformer encoder sub-block naturally have the form of such a task representation when given the context trajectory $c_t^i$ of the form (18) as input at a particular time point $t$: Reformulating the self-attention softmax term (21) for a particular transition $o_t$ and the multiplication with the corresponding value vector $v_t$, one yields representations for the coefficients $\alpha$ and the linear function $f$ in (16) - the former through rewriting the softmax term, the latter by a sum over entries of the value vector. For the rather technical full derivation of that MDP representation property, the interested reader is referred to the original publication (Melo, 2022). For the scope of this work, it is sufficient to conclude that it creates the implicit objective to "learn to make a distinction among the tasks in the embedding space" (Melo, 2022), i.e., the implicit goal to learn how to identify tasks through the collected experience. In fact, (Melo, 2022) additionally prove that any Transformer self-attention layer minimizes the task uncertainty based on the current context $c_t^i$, i.e., that every Transformer sub-block finds the best representation of the gathered information. In each Transformer sub-block, the previous representation of $c_t^i$ is recombined through the dense layers before the next self-attention layer persists in the form of a task representation. In this way, the representation of $c_t^i$ is refined accross episodes, which "resembles a memory reinstatement operation" (Melo, 2022), i.e., an operation to reactivate long-term memory in order to identify the current task.

**Meta-Training and Meta-Testing**

In RL, transformers are often unstable during training (Hu et al., 2024), especially in the beginning (Melo, 2022). And while more sophisticated algorithms like AMAGO (Grigsby et al., 2023) stabilize training by incorporating off-policy learning into their Transformer architecture, TrMRL addresses this problem right at the beginning of meta-training by utilizing T-Fixup initialization (Huang et al., 2020). This initialization scheme is especially designed to avoid learning rate variation and layer normalization during early training, and the ablation studies of (Melo, 2022) particularly show its usefulness. Besides that, the meta-training and meta-testing schemes of TrMRL directly derive from the RL$^2$ schemes provided in Section 3.2.

**Performance Analysis**

As implicitly shown in the performance analysis paragraphs of the previous subsections, TrMRL outperforms MAML, VariBAD, and, most importantly, RL$^2$ in the MuJoCo environment w.r.t. sample-efficiency, adaptation speed and particularly OOD performance (see Melo (2022), Figures 4, 5 and 6). This superior performance is even more significant in the more sophisticated Meta-World environment (see Melo (2022), Figures 4, 5 and 6), where it achieves equal or higher success rates than the other landmarks in all except the first episodes (in first episode, VariBAD is slightly better). The only exceptions are MuJoCo tasks with high task uncertainty, e.g., Meta-World-ML1-Push-v2, where VariBAD oscilates between a success rate of 0.5 and 0.8, while TrMRL remains stable at a value of 0.5 (see Melo (2022), Figure 4). This motivates combining a Transformer-based meta-RL agent with Bayesian inference such as in PSBL (Xu et al., 2024), or ADA.

## 3.5 The Adaptive Agent

Transformers still struggle with "credit assignment" (Ni et al., 2023), i.e., with assigning current actions to rewards later on in the planning horizon - a problem that even increases when the data complexity is high (Ni et al., 2023). This particularly motivates the use of model-based RL, where the agent can base its decisions on the predictions of the dynamics model. The only landmark algorithm on the development path towards ADA utilizing model-based RL for inner learning is VariBAD (Section 3.3). It additionally models task uncertainty into its paradigm and, hence, (Melo, 2022) already identify the possibilities of combining the VariBAD paradigm with a transformer architecture. The Adaptive Agent is this transformer-based modification of VariBAD. It combines a large Transformer architecture with self-supervised learning techniques in order to develop a generalist agent applicable to a vast number of "downstream tasks", i.e., test tasks significantly different from the meta-training tasks - perhaps even OOD. In this respect, it builds upon other, similar approaches aiming for a generalist agent, i.e., RL foundation models, like GATO (Reed et al., 2022) or the work of (Team et al., 2021).

Analogous to VariBAD, ADA consists of a policy $\pi_\theta$, a decoder $g_\psi$, and a context encoder $f_\varphi$ whose output is the variational distribution $p_\varphi(z|c_t^i)$ representing task uncertainty. However, the ADA paradigm modifies these components as follows:

- Prior to the Transformer architecture, a ResNet architecture preprocesses the visual 3D observations received from the XLand environment.

- The memory-based context encoder is a Transformer architecture; [14] more precisely, the Transformer-XL architecture (Dai et al., 2019), a demonstrably more efficient and powerful modification of the Vanilla Transformer whose architecture allows it to model long-range dependencies.

- The architecture following the Transformer-XL mimics the Muesli algorithm (Hessel et al., 2021), a computationally efficient, demonstrably powerful RL approach that combines model-based and model-free RL. This requires a critic network estimating the value (6) of the current state as well as a dynamics model for planning. The latter predicts policy and critic network outputs along with future rewards and transitions.

---

[14]Note that theoretically any Transformer architecture can be used that is suitable for RL.

In addition to these modifications, the number $K$ of adaptation episodes varies (i.e., $K = \{1,\ 2,\ \ldots,\ 6\}$). Hence, $k$ is an additional input to the context encoder.

### Meta-Training and Meta-Testing

In contrast to TrMRL, the ADA meta-training scheme does not consist of a particular initialization scheme. Instead, layer normalization and gating techniques stabilize the transformer during training. However, as those are rather detailed technical details, the interested reader is referred to the original work (Team et al., 2023). Besides that, the VariBAD meta-training scheme is extended by two self-supervised learning techniques: 1) An automatic curriculum (ACL) selecting tasks on the frontier of the current agent's capabilities for efficient learning, and 2) distillation for kickstarting the training process. The subsequent subsections describe both these techniques in more detail. But before, the following paragraphs summarize the performance analysis provided in the original work (Team et al., 2023).

### Performance Analysis

The Adaptive Agent achieves human-like few- and zero-shot performance (in terms of generalization and adaptation speed) for single and multi-agent tasks on an even more complex and dynamic extension of the XLAND environment (Team et al., 2021), a "vast open-ended task space with sparse rewards" for single and multi-agent tasks. In multi-agent downstream tasks, the different agents clearly share labor and actively cooperate to improve their reward, while in single-agent tasks it is able to use one single episode of expert demonstrations in order to significantly improve performance (Team et al., 2023). This makes ADA superior to the work of (Team et al., 2021) and other generalist agents like GATO (Reed et al., 2022) which exclusively learn from demonstrations. However, ADA is not generally superior to human-level performance, as there are single- and multi-agent downstream tasks neither humans nor ADA are able to solve.

ADA's transformer memory length, i.e., the length of the context $c$ used as model input, as well as its model and task pool sizes, scale demonstrably well, particularly when being scaled together with each other (Team et al., 2023). For example, increasing the task pool, i.e., sampling a larger number of distinct tasks from a task distribution with more complex tasks, always has a positive effect on the agent's median performance (i.e., the median of the performance over all tasks), especially in the few-shot setting. But this effect is even bigger when model size or memory also increase. The same holds for the model size and the transformer memory length: larger models as well as models with longer context windows obtain better median performance in any $K$-shot setting, particularly when the task pool also increases. On a log-log plot, the effect of increasing the model or context window size additionally scales roughly linearly with the number $K$ of shots. But only if the task pool is sufficiently large. Otherwise the model overfits the small task pool. All these findings also hold for the 20th quantile i.e., in the 20,This indicates a more stable training and underpins the work of (Bommasani et al., 2021) which identify scaling as the key factor of what makes foundation models so powerful.

Although ADA is capable of handling varying context lengths, it, like most RL Transformers, assumes the dimension of observations to be constant throughout tasks. However, as ADA particularly utilizes an additional observation encoder prior to its transformer architecture, it suffices to exchange only that encoder when observation spaces change. This motivates AMAGO (Grigsby et al., 2023), a more flexible approach that particularly utilizes an observation encoder mapping each context to a fixed-sized representation, so that the encoder is the only required architectural change across experiments. This way, AMAGO is particularly designed to incorporate off-policy actor-critic RL in combination with distinct, dynamic, long-term contexts, so that it is applicable for multi-task RL.

Since approaches like AMAGO, GATO, and ADA clearly are foundation models, they are computationally expensive during meta-training (Shala et al., 2024b), which motivates hierarchical transformer architectures like HTrMRL (Shala et al., 2024a) or ECET (Shala et al., 2024b). They use additional self-attention sub-blocks to process cross-episode data in order to further increase sample efficiency and reduce model complexity. In contrast to the intra-episode self-attention sub-blocks utilized in TrMRL, AMAGO, and ADA, these cross-episode blocks set whole episodes of experience in context to each other, so that the context mechanism of self-attention is also leveraged at episode level. Among other model reduction techniques like

distillation, this is a promising direction for future work in the field of foundation models and generalist agents.

### 3.5.1 Automated Curriculum Learning:

Automated Curriculum Learning (ACL) leverages the idea from human learning that training tasks should not be too hard or too easy for the current level of learning. It originates from curriculum learning (CL), (Dansereau, 1978), where curricula are hand-crafted to guide the learning process of a standard learner (e.g., a neural network) on a single task. Such curricula were successfully applied in several supervised learning tasks like NLP, computer vision, medicine, Neural Architecture Search, and RL (see e.g., Wang et al. (2022b), Soviany et al. (2022), Gupta et al. (2021b), for a detailed overview). The main idea of ACL is to automatically prioritize tasks at the frontier of the agent's capabilities during training, instead of manually selecting them. This ensures that the agent is consistently challenged and can progressively improve its skills (Portelas et al., 2020), so that sample efficiency and generalization performance can be significantly improved (Dennis et al., 2021), (Jiang et al., 2021), (Jabri et al., 2019), (Wang et al., 2022b).

There are several ACL methods (see e.g., Portelas et al. (2020) for a detailed overview). One of the earliest approaches is Teacher-Student Learning (Matiisen et al., 2020), where a second model (the teacher) is simply used to select subtasks for the standard learner (the student) during training. For meta-RL, there also exist various techniques, e.g.,

- Simple, but quite static methods like domain randomization (e.g., Volpi et al. (2021)), where environments are generated randomly, but independently of the current policy's capabilities.

- More flexible approaches like min-max adversarial (e.g., Volpi et al. (2018)), where environments are created adversarially, i.e., to challenge the agent as much as possible.

- Sophisticated paradigms like min-max regret, e.g., Protagonist Antagonist Induced Regret Environment Design (Dennis et al., 2021), where an adversary aims to maximize the regret between the protagonist (the agent) and its antagonist (an opponent). The latter is assumed to be optimal.

For min-max regret, the regret is defined as the difference between the value of the protagonist's and the antagonist's action. This way, the adversary selects the simplest tasks the protagonist is not yet able to solve (Dennis et al., 2021). This makes min-max-regret superior to min-max-adversarial, which, due to its objective, tends to generate tasks too difficult for the agent or even unsolvable (Dennis et al., 2021). However, ADA leverages and compares two other ACL methods during meta-training:

1. NOOB-Filtering (Team et al., 2021) maintains a control policy that takes no actions. After rolling out ADA and the control policy for ten episodes in a new task, it is selected for meta-training if: ADA's performance is neither too good nor too bad, the variance among trials is sufficiently high across episodes to indicate proper learning, and The control policy performs poorly enough to indicate the relevance of proper decision making.

2. Robust Prioritized Level Replay (PLR) (Jiang et al., 2021) maintains a "level buffer" of tasks with high learning potential. It samples tasks from that buffer with a probability of $p$, and otherwise samples a new task from the task distribution $p(T)$.

   The task's regret estimates its learning potential. A task is added to the level buffer, if its regret is higher than those of the other level buffer tasks. This constantly refines the level buffer. In ADA, several fitness metrics like policy and critic losses approximate the task regret.

In comparison to meta-training with uniformly sampled tasks, both no-op filtering and PLR, improve ADA's sample efficiency and generalization as well as its few- and zero-shot performance (Team et al., 2023). Additionally, PLR slightly outperforms NOOB filtering, especially when the number $K$ of adaptation episodes is higher. However, any other approach from the ones described above might work just as well.

### 3.5.2 Distillation

The notion of distillation was first proposed by (Hinton et al., 2015). It generally refers to model compression techniques transferring knowledge from a large, pre-trained teacher model $T$ [15] to a smaller, more efficient student model $S$ (Rusu et al., 2016). Hence, distillation can be viewed as a transfer learning approach (Mansourian et al., 2025). Classically, the student is trained to learn from the teacher's demonstrations, i.e., it learns to predict the teacher's softmax output distribution (in the discrete case) that is smoothed by choosing a sufficiently high softmax temperature parameter in order to yield soft targets (Rusu et al., 2016), (Mansourian et al., 2025), (Hinton et al., 2015). In RL, one of the best-known distillation approaches is policy distillation (Rusu et al., 2016) (also known as imitation learning), where the student policy is trained to imitate the teacher's action distribution. Other widely used approaches include value function distillation and dynamic reward-guided distillation (see Xu et al. (2025) for further details).

Generally, the two main questions in selecting a knowledge distillation scheme are: 1) What knowledge to transfer and 2) which architectures to choose for teacher and student models (Mansourian et al., 2025), (Gou et al., 2021). ADA uses dynamic reward-guided distillation, where the student policy is guided by the teacher policy rather than particularly mimicking it. It contains an additional distillation loss throughout the first four billion meta-updates that consists of the KL-divergence between student and teacher policy action distributions, as well as a $L^2$-regularization term like in (Schmitt et al., 2018). This way, the student policy can also explore states different from those visited by the corresponding teacher model, which was trained from scratch with a model size of 23 million parameters.

For comparison, (Team et al., 2023) train four models: A large (256 million parameters) and a small (23 million parameters) model, each with and without distillation. They show the larger distilled model to strongly outperform the smaller one as well as the large ordinary model, while the smaller ordinary model outperforms the larger one. This not only indicates a significant positive effect of distillation on model performance, but also highlights its necessity for large-scale (foundation) models.

## 4 Discussion

Meta-learning describes the paradigm of learning how to learn. However, given the vast array of potential tasks, the question naturally arises: What general knowledge should be acquired? Earlier algorithms primarily focused on meta-learning the skill to make decisions in RL environments by pre-training model weights (MAML), and by basing the decisions on the current context (PEARL and RL$^2$) or the current context-based belief about which task they are acting in (VariBAD) with manually designed architecture modifications. Transformer-based agents are meta-learners by design. Although they were not specifically designed to be meta-learners, these models learn how to learn just by emergence.

The path towards general intelligence often looks like this (Bommasani et al., 2021): While earlier approaches are hand-designed for particular purposes, later ones yield more general, unsupervised, and unguided intelligence, learning different skills emergently. The consequence is a shift towards "homogeneity", i.e., the use of a single model architecture for several different tasks or task distributions (Bommasani et al., 2021). In other words, as soon as a model architecture generalizes better, it is used for more general purposes. The path towards the Adaptive Agent presented in Section 3 exemplifies this development. Even within the transformer-based development path, one can clearly observe this shift towards homogeneity: While early transformer-based meta-RL architectures like TrMRL require hand-designed architectures for different observation spaces, generalist agents like ADA and AMAGO abstract this problem by observation-space-specific encoder blocks. Observing this development scheme (as outlined by (Bommasani et al., 2021) and (Clune, 2020)), suggests how future developments might look like: As, from a theoretical point of view, any skill can be meta-learned, meta-learning does not have to solely focus on task-specific adaptation. The easiest example is meta-learning hyperparameters, - an approach that appeared quite early in the meta-learning timeline, with algorithms such as $\alpha$-MAML (Behl et al., 2019) or PEARL (Rakelly et al., 2019). But even far more complicated skills such as neural architecture search, curriculum design or pre-designing popula-

---

[15]It is also possible to use multiple teachers for different subtasks (Schmitt et al., 2018). However, these teachers can be seen as one teacher model consisting of different task-specific components.

tions of evolutionary algorithms, can be meta-learned. The last of the following subsections discusses such approaches and sets them in context to the path towards general intelligence.

### Relevance of Meta-Learning

With the shift from manually designed, relatively simple to understand meta-learning algorithms like MAML or PEARL to largely scaled, blackbox foundation models like GATO (Reed et al., 2022), ADA (Team et al., 2023), or AMAGO (Grigsby et al., 2023), the question arises whether the paradigm of meta-learning is actually still of relevance. Large-scale foundation models from companies like DeepMind, OpenAI, or Meta can be applied to a vast amount of in- and out-of-distribution downstream tasks of different types - as e.g., done for the LAMA model in (Rentschler & Roberts, 2025) for RL tasks - without even thinking about data quality, distribution shifts, or the meta-training paradigm. However, such foundation models require a vast amount of computational power, even for simple downstream rollouts without model updates. During training, they necessitate scaling of their model size, the task pool, and the task complexity (Team et al., 2023), (Bommasani et al., 2021). Such an amount of data and computation power is not available for all researchers, companies and people - even when techniques like distillation and ACL decrease the model size and boost the training progress. Moreover, in recent years there has been a significant shift from publicly available models with revealed source code trained on open-source data towards secretly training models of unknown sizes and shapes on largely collected datasets that are hidden from the public - and, as such, hidden from public control and revision. But how models behave highly depends on the data they were trained on, so that this privatization of model training comes with a high risk of unexpected, harmful behaviour (Bommasani et al., 2021), along with the societal risk of an increasing gap between institutions with a high amount of computational power and data and others who cannot utilize large resources.

Modern research can, regardless of the available computation power, focus on different niches of application (Togelius & Yannakakis, 2024) as well as on developing more efficient architectures like hierarchy transformers (Shala et al., 2024a), (Shala et al., 2024b), or on gaining a better understanding of blackbox models, their behaviour, the reasons for their failures, and the societal and ethical impact their application has (Togelius & Yannakakis, 2024). The latter can support the development of even better general problem solvers (Clune, 2020), so that collaboration between university researchers and Big Tech companies is probably beneficial for both (Togelius & Yannakakis, 2024), (Bommasani et al., 2021).

### The Need for Specialized Meta-Learners

In application, agents must often be deployed on small edge devices with a very limited amount of computational power that does not allow for generalist agents but might benefit from fast adapting meta-models. Such problems likely cover only a very narrow, low-dimensional task distribution that does not require generalist agents. In other words, largely scaled transformer-based architectures often go far beyond what is needed to solve a particular problem. This likely explains why many meta-learning applications still rely on simple, sample-efficient gradient-based algorithms like MAML, even though they often have poorer OOD performance (see e.g., the applications for medicine Tian (2024), Alsaleh et al. (2024), Tian et al. (2024), Ranaweera & Pathirana (2024), Naren et al. (2021), biomass energy production (Zhang et al., 2025), or fault diagnosis (Lin et al., 2023)). However, understanding the advantages and drawbacks of the various components of the landmark algorithms presented in the previous section can support applied researchers in manually designing the best fitting meta-learner for their particular problem. For example, autonomous driving requires extremely fast adaptation, very good zero-shot performance and high robustness and reliability, while sample efficiency does not really matter, especially during training. In such an application, gradient-based algorithms are most likely the wrong choice, while memory-based Bayesian inference learners like VariBAD or ADA are much more promising. In contrast, medical applications typically have a very limited amount of data available per patient, so that they particularly require high sample efficiency. For such applications, gradient-based meta-learners are a good first choice, as they are simple and more sample-efficient. This especially holds true for off-policy meta-RL algorithms like PEARL. To assist the reader in selecting a meta-learning algorithm for application, Table 1 provides an overview of the main advantages and drawbacks of the developments discussed along the timeline of the previous section.

Table 1: Advantages and disadvantages of the different components introduced along with the landmarks in Section 3. The last column lists algorithms containing the respective component.

| Component | Advantages | Drawbacks | Algos |
|---|---|---|---|
| Gradient-based | • Easy to implement
• Usable for meta-learning and meta-RL
• Many open-source repos available | • Weak generalization and OOD performance • Only PG methods for meta-RL | MAML, FO-MAML |
| Context-based | • Dynamically adjust to tasks without model training
• Benefitial in sparse-reward tasks
• Implicit goal of base Bayes-optimal behaviour | • Highly dependent on the quality of context | CAVIA, DREAM, MAESN, Meta-Cure, PEARL, RL$^2$ |
| RNN-based | • Sequence processing, context-based by design
• Simple memory-based architecture | • Poor asymptotic performance
• Sensitive to architecture, context length, and RL algorithm choice | RL$^2$, VariBAD |
| Bayesian Inference | • Explicit uncertainty quantification
• Tailored for Bayes-optimal behaviour | • Complex implementation
• Often exhibit unstable learning | MAESN, Meta-Cure, PEARL, VariBAD, MELTS |
| Model-based RL on Task-Level | • Decisions can be based on the dynamics model's predictions; better long-horizon planing | • Higher architectural and computational complexity | VariBAD, ADA |
| Transformer-based | • Powerful contextualization
• Meta-learner per design
• Very high sample efficiency and adaptation speed
• Particularly high OOD performance | • Require vast amounts of data and computational resources for (meta-)training
• Perform worse on tasks with high task uncertainty | TrMRL, ADA, AMAGO, GATO, ECET |
| ACL | • Improved meta-training efficiency | • May be costly to implement | MELTS, ADA |
| Distillation | • Reduce model size
• Stabilize initial meta-training | • Potential loss of performance | HyperX, ADA |
| Generalist Agents | • Can handle a wide variety of tasks | • High computation cost even for rollouts | ADA, AMAGO, GATO, ECET |

**Comparability**

There is a lack of comparability among the landmark algorithms presented in the timeline of the previous section. The utilized performance measures differ between the different works and are often not clearly defined, an issue this work aims to address by defining general performance measures in Section 2.3. In addition, different works utilize different benchmarks with different properties and of different complexity. Thereby, the development of those benchmarks follows the development of meta-RL algorithms itself: Early landmarks like MAML, RL$^2$, or PEARL were mainly tested on simple grid-world environments, the Atari benchmark (Bellemare et al., 2013), or the MuJoCo engine (Todorov et al., 2012). But (Fakoor et al., 2020) identified that those benchmarks are too simple to evaluate actual adaptation, which motivates more generalistic, complex, and sparsely rewarded environments like Meta-World (Yu et al., 2020), Crafter (Hafner, 2021) or XLand (Team et al., 2021). This lack of homogeneity makes it difficult for researchers and developers

to compare different algorithms and modifications. However, this is a general problem of ongoing research and development and can hence, only be tackled by several empirical studies.

**The Three Pillars of the Path Towards General Intelligence**

The development path towards human-like or even superior generally intelligent agents consists of three pillars (Clune, 2020):

- Meta-learning algorithms that can be viewed as the "evolution of intelligence" itself,

- meta-NAS (Pereira, 2024), (Hospedales et al., 2022) creating the "physical body" of the gained "intelligence," and

- the generation of more meaningful, complex, and challenging environments that function as the "world" the "physical body" of the learned "intelligence" is acting in.

While research in the meta-learning community primarily focused on the pillar of learning algorithms and, secondly, on meta-NAS approaches, meta-environment research is very sparse (Clune, 2020). However, as the growing complexities and capabilities of recent developments come with the necessity of a further increase in complexity for benchmark environments, the attention most recently shifts towards this research field by combining meta-RL with self-supervised learning techniques. This is reflected in the ADA paradigm, where initial learning is guided by a teacher model (Distillation) and an automated curriculum (ACL) selecting tasks on the frontiers of the agent's capabilities. The latter is, however, a skill that can be meta-learned, so that a (teacher) model learns to create a personalized curriculum for any student model (Portelas et al., 2021), (Xu et al., 2023). The corresponding Meta-ACL paradigm is inspired by classroom scenarios, where a teacher teaches several students at once, ideally with respect to their personal capabilities and special needs.

Additionally, meta-design of environments can be inspired by examining the evolution on Earth (Clune, 2020). Instead of developing algorithms generating open-ended environments like XLand (Team et al., 2021) and meaningful rewards, one can aim for agents that explore out of curiosity (Alet et al., 2020), i.e., on their own, or through intrinsic reward (see e.g., Zhang et al. (2021)) in environments providing a high variety of different niches to adapt to. The former corresponds to techniques like novelty search (Lehman & Stanley, 2011), while algorithms like quality diversity (Pugh et al., 2016) can be utilized to generate high-variety environments.

Considering the current work of Big Tech companies, a very likely future path of meta-learning and foundation model development will be towards combining these pillars. The most recent landmark of that kind is the Evolution Transformer (Lange et al., 2024) of DeepMind. It combines the pillars of meta-learning and meta-NAS by learning how to design the population of RL agents (individuals) for evolutionary RL algorithms. In other words, on top of the evolutionary approach, the Evolution Transformer learns how the population of agents (breed of creatures) must be designed to have the highest chance of survival as a whole, i.e., it focuses on the continued existence of the species rather than on maximization the performance of single individuals.

Combining the different pillars results, along with a potential increase in model capability, in several potential ethical, societal, and economic concerns that are very difficult to predict. When models are "taught to play god" - even in a very limited world like the currently existing environments - it is essential to monitor their behaviour, the basis on which they make their decisions and the harm they can potentially cause from different views like law (Chan et al., 2024), ethics, economy and various others. In this respect, the future development might be examined and analyzed analogously to that of foundation models in (Bommasani et al., 2021), where a large group of various researchers of several domains and backgrounds collaborate in order to identify potential risks, challenges and benefits of the current shift towards large-scale foundation models.

## 5   Conclusion

This comprehensive survey provides the timeline of meta-RL developments from foundational algorithms like MAML and $RL^2$ to ADA, a large-scale generalist agent. Through detailed mathematical formalizations

of meta-learning and meta-RL paradigms, this work addressees the lack of detailed formalism within the literature and lays the groundwork for understanding the paradigms and training schemes of the landmark algorithms presented along that timeline. Based on that formalized knowledge, it highlights the paradigm shift towards transformer architectures and large-scale foundation models through the usage of self-supervised learning techniques like distillation and automated curriculum learning. Looking ahead, this survey examines the three pillars of general intelligence and identifies a trend towards the automated generation of environments through meta-learning of self-supervision and evolutionary approaches. Besides offering valuable insights, this highlights the constantly growing need for collaborative, interdisciplinary teams of researchers that permanently analyze the behaviour of generalist agents and their ethical implications for our society.

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

## A  Bayesian Reinforcement Learning

Bayesian Reinforcement Learning (BRL) provides a sophisticated approach to tackle the exploration/exploitation dilemma by allowing agents to quantify uncertainty. It is the groundwork for the task-representation algorithms presented in sections C and 3.3. The main idea of BRL is to leverage the gathered information of the agent together with Bayesian methods in order to enable the agent to adapt its strategy as more information becomes available. For this purpose, one extends the notion of a MDP of the form (5) to that of a Bayes-Adaptive MDP (BAMDP)[16], i.e., a tuple

$$M' := (\mathbb{A}, \mathbb{S}', R', \gamma, \mathbb{T}', \mu', h'),$$ (17)

with a hyperstate-space $\mathbb{S}' = \mathbb{S} \times B$ as the extension of the original state space $\mathbb{S}$ by the belief space $B$ capturing the possible parameters of a model of the environment dynamics $\mathbb{T}$ and $R$[17]. The augmented transition and reward functions, as well as the initial distribution and horizon, are defined analogously to the hyper-state space (see Zintgraf et al. (2020), Ghavamzadeh et al. (2015) for further details). In terms of Example 2, this means, for example, to capture a belief of the current slipperiness of the track while collecting more experience to make this belief more precise.

The current belief $b_t := p_t = p(R', \mathbb{T}'|c_t) \in B$ is defined as the posterior of the dynamics model parameters given the current experience (also named context)

$$c_t := \{s_i, a_i, r_{i+1}, s_{i+1}\}_{i=0}^t$$ (18)

and a prior distribution $p_0 = p(R', \mathbb{T}')$ over the unknown reward and transition functions $\mathbb{T}$ and $R$. The prior is often chosen as a normal distribution, while one computes the posterior $p_t$ according to the Bayesian rule after collecting experience $c_t$:

$$p_t = p(R, T|c_t) = \frac{p(c_t|R, T)p(R, T)}{\int p(c_t|x)p(x)dx}.$$ (19)

The posterior encodes the agent's uncertainty about model environment parameters. However, its computation is generally intractable so that the respective BRL algorithm must approximate it accordingly. Since the agent aims to select the best possible action in each time step based on the current belief, while simultaneously refining the posterior belief whenever new information becomes available, it implicitly trades-off exploration and exploitation this way.

One calls agents Bayes-optimal, if they optimally trade-off exploration and exploitation. Throughout early learning, the value (6) of the corresponding Bayes-optimal policy is typically lower than that of the optimal policy $\pi^*$, since the agent must, at first, explore the environment before finding the optimal dynamics model parameters. In Example 2, the Bayes-optimal behaviour initially requires driving and braking a few times to figure out how slippery the track is, although this increases the racing time in the first training episode.

---

[16]In (Ghavamzadeh et al., 2015) it is done for discrete state and action spaces, but it is straight forward to adapt this for continuous spaces

[17]As all BRL-based algorithms presented in Section 3 use model-based approaches, model-free BRL is not discussed in this work.

# B    Terminology

In the literature, meta-learning is often confused with the notions of transfer learning (TL) and multi-task learning (MTL). Although these concepts share quite some similarities - there are even algorithms exhibiting overlapping characteristics among them (Upadhyay, 2023) - the confusion mainly arises from a lack of clear definition. Therefore, the following paragraphs broadly introduce TL and MTL focusing on a clearly defined paradigm. For a more detailed consideration, the reader is, nevertheless, referred to other works like (Upadhyay, 2023) particularly focusing on meta-learning, MTL, TL and their overlaps.

## B.1    Transfer Learning

The significance of the concept of Transfer Learning notably heightened following its integration with deep neural networks. Landmark advancements in computer vision, exemplified by architectures such as AlexNet (Krizhevsky et al., 2012), GoogleNet (Szegedy et al., 2015), and EfficientNet (Tan & Le, 2019), are pre-trained on the extensive ImageNet dataset before employing them to tasks such as image classification, segmentation, and object detection. These models demonstrated substantial improvements over traditional machine learning algorithms in terms of adaptability, sample efficiency, and few-shot performance. Motivated by these achievements, transfer learning has also been effectively applied in various other domains, including speech recognition (e.g. Wav2Vec Schneider et al. (2019)), healthcare applications (Esteva et al., 2017), and reinforcement learning (RL). The latter itself distributes into various applications as knowledge transfer is used to close the gap between simulation and reality in robotics (Yu et al., 2019), learn from demonstrations to incorporate expert knowledge (Ravichandar et al., 2020), (Sosa-Ceron et al., 2022), or adapt to new rules and gaming mechanics in game play using reward shaping (OpenAI et al., 2019). All these various sub fields of TL and transfer RL present distinct challenges and are hence areas of ongoing research. Interested readers are thus encouraged to explore specific surveys on TL (Niu et al., 2020), (Zhuang et al., 2021), (Tan et al., 2018), (Panigrahi et al., 2021), and transfer RL (Zhu et al., 2023), (Zhao et al., 2020) for deeper insights.

### Transfer Learning Paradigm

The main idea of TL is to extensively pre-train a model $F_\theta$ on a source task $T_1 := \{\mathcal{L}_1, \mathcal{X}_1, \mu, \mathbb{T}_1, h\}$ and transfer the acquired knowledge by fine-tuning $f_\theta$ on a related target task $T_2 := \{\mathcal{L}_2, \mathcal{X}_2, \mu, \mathbb{T}_2, h\}$. Mirroring human learning, this approach leverages knowledge from pre-training to enhance few-shot performance and learning speed on the target task $T_2$. For example, one can extensively practice inline skating in the summer to prepare for skiing when it snows for only a few days during winter. Therefore, the source task $T_1$ typically contains abundant, well-labeled data $\mathcal{X}_1$, while the target task's data $\mathcal{X}_2$ often is sparse, of low quality or expensive to obtain. Additionally, the goal encoded by $\mathcal{L}_1$ can differ from that represented by the loss function $\mathcal{L}_1$ or there may be a change in domain dynamics $\mathbb{T}$ that needs to be learned quickly. The latter is of particular importance in RL problems, where changes in environment dynamics can cause severe issues; for example when an agent is supposed to drive a car and the terrain becomes significantly more slippery. A change in the loss function can, however, become problematic if the loss $\mathcal{L}_2$ is considerably more computationally expensive to calculate as this slows down learning.

One can also define the TL paradigm with the possibility of several source tasks. However, this is rather a technical difference as one can define $T_1 = \bigcup_{i=1}^{N} T_1^i$, where $T_1^i$ are the different source tasks. If, however, more than one target task exists, one typically uses the notion "downstream tasks". This is the case in more recent developments, where TL, along with a vast amount of data and self-supervised learning approaches, led to much more powerful and broadly applicable models, namely foundation models.

### Foundation Models

Foundation models serve as a common basis from which many task-specific models are built through adaptation (Bommasani et al., 2021), i.e. they are pre-trained by a vast amount of source tasks as described above, before being fine-tuned to different downstream tasks. As a consequence, foundation models are inherently incomplete: they rather serve as a "foundation" for fine-tuning to various applications than as static general

problem solvers. This way, the adaptation of foundation models follows the meta-testing paradigm described in Section 2.1, while their training is that of a TL algorithm not following the meta-training paradigm of Section 2.1. As they are meant to be trained on a distribution $p(T)$ of training tasks rather than one single source task before being fine-tuned to a (potentially different) distribution of downstream tasks $p_{\text{test}}(T)$, they can be viewed as the "meta-level version" of TL.

TL is the key technique that makes foundation models possible, but scale is what makes them so powerful (Bommasani et al., 2021). This fact probably arises from the universal approximator property of neural networks: larger models can represent more complex functions of higher dimension, although they also tend to overfit more easily. The latter is the reason why besides model size, the data (e.g., the task pool in meta-learning) is also required to scale (Team et al., 2023), (Bommasani et al., 2021). However, as the amount of data increases, the data quality can not necessarily be maintained. Instead, self-supervised learning techniques (Gui et al., 2024), (Ericsson et al., 2022) are used to learn patterns in available, but potentially unlabeled, data, so that foundation models are often defined as "large-scale transfer-learning models pre-trained via self-supervised learning" (Bommasani et al., 2021).

There was a paradigm shift towards foundation models that get fine-tuned for downstream tasks within the recent years (Bommasani et al., 2021). For example, almost all state-of-the-art NLP models are now adapted from one of a few foundation models, e.g., Bert (Devlin et al., 2019), GPT3 (Brown et al., 2020), GPT4 (OpenAI et al., 2024), LLAMA (Touvron et al., 2023). Similarly, foundation models were developed for various applications like agriculture (Yin et al., 2025), medicine (Zhang & Metaxas, 2024), (Liang et al., 2025), (Moor et al., 2023), or robotics (Firoozi et al., 2025). In RL, foundation models have been used for reward engineering (Ye et al., 2024), (Wang et al., 2024b), e.g., to generate rewards from only text description and the visual observations yielded from the environment (Wang et al., 2024b). Additionally, (Rentschler & Roberts, 2025) successfully fine-tune the LLAMA (Touvron et al., 2023) foundation model on RL downstream tasks without RL-specific training with remarkable results. As a consequence, several works combine NLP and RL paradigms into RL from Human Feedback (RLHF), which is itself an ongoing and vast research field. We refer interested readers to particular RLHF surveys like (Kaufmann et al., 2023), (Casper et al., 2023), (Chaudhari et al., 2025).

## B.2   Multi-Task Learning

The concept of Multi-Task Learning (MTL) was first introduced by (MTL) as the formalization of the idea of training models on multiple tasks simultaneously. The main idea is to train one single model among several similar tasks simultaneously in order to find valuable, general feature representations that can be leveraged to improve task-specific performances.

Applying the paradigm of MTL to different neural network architectures like Transformers (Torbarina et al., 2024) or Bayesian networks (Lazaric & Ghavamzadeh, 2010), ..., led to remarkable successes in natural language processing (Chen et al., 2024), neural architecture search (Gao et al., 2020), (Pasunuru & Bansal, 2019), segmentation (Bischke et al., 2019), (Zhou et al., 2021), (Tang et al., 2023), medicine (Bi et al., 2008), (Zhang et al., 2023b), (Zhou et al., 2021), and robotics (Arcari et al., 2023), (Gupta et al., 2021a), (Deisenroth et al., 2014), to list only a few. We, however, refer interested readers to MTL-focused surveys like (Zhang & Yang, 2022), (Zhang & Yang, 2018), (Yu et al., 2024) or Multi-Task-RL (MTRL) specific works like (Vithayathil Varghese & Mahmoud, 2020) or (D'Eramo et al., 2024) for a broader overview of MTL and MTRL algorithms. This section rather focuses on MTL and MTRL paradigms to help readers distinguish them from TL and meta-learning or transfer-RL and meta-RL respectively.

### Multi-Task Learning Paradigm

Formally, in MTL a model is jointly trained to solve a given number $N$ of tasks $(T_1, T_2, \ldots, T_n)$ sharing some common structure. The corresponding training process mimics standard learning on task-specific level, but with parameters being shared throughout the tasks. A meta-level does not exist and hence there is no two-staged training process. Instead, the MTL paradigm aims to solve all $N$ tasks equally well. Figure 6

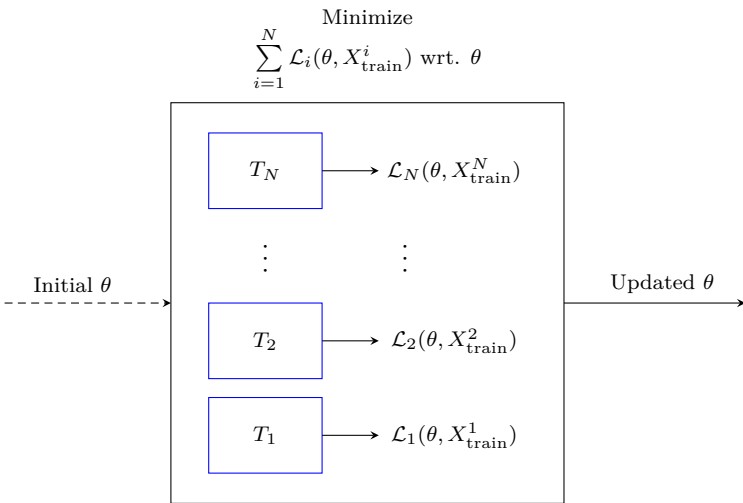

Figure 6: General Multi-Task-Learning Paradigm:

visualizes the resulting optimization problem. It can be formalized as

$$\text{Minimize} \quad \sum_{i=1}^{N} \mathcal{L}_i(\theta_i, X_{\text{train}}^i) \quad \text{wrt. } \theta \tag{20}$$

where $\theta_i \in \mathbb{R}^{d_i}$ denotes the task-specific parameters reserved for each individual task.

The task specific parts of the model typically are task-specific heads with one or more layers, so that the model parameters $\theta$ consist of task-specific parameters $\theta_i$ for each task $T_i$ as well as common knowledge denoted by $\varphi$, but all in one single model. The performance of the MTL model is evaluated on task-level only, i.e. on the task-specific test sets $X_{\text{test}}^i$. A meta-test on unseen tasks does not take place.

In contrast to the meta-learning paradigm described in Section 2.1, the MTL paradigm does not treat the parameters $\theta_i$ of each task as individual parameter sets. They are rather considered as the different parts of one bigger set of parameters $\theta = \{\theta_0, \theta_1, \ldots, \theta_N\}$ with $\theta_0 \in \mathbb{R}^{d_0}$ denoting the common knowledge over all tasks. A deep neural network can, for example, learn some feature representations shared between all different tasks to extract the relevant information of its data before fine-tuning to the respective task. Such a NN theoretically consists of different blocks: A general part encoding all the information shared between tasks, and a smaller head for each individual task.

**Multi Task Reinforcement Learning**

In Multi-Task Reinforcement Learning (MTRL) one develops a policy for several similar but different tasks at once. Analogous to the general MTL paradigm, the common knowledge is about how to process the information the current state provides, which action has which effect on the environment and what are general dynamics shared among tasks. At the same time, rewards and transitions differ between the different MDPs, what requires for task-specific heads to decide for an action out of the (commonly) processed state information. Since the notion of a policy is rather a theoretical concept, this means the same neural network can function as the agent in the different MDPs with a task-specific head for each MDP. A second part of potentially common knowledge is task identification. However, compared to the more general meta-RL paradigm, this is rather simple: The model always faces the same $N$ tasks. Hence, most algorithms simply use one-hot encoding.

In the literature, MTRL is often confused with Multi-Agent learning - where multiple agents act within the same environment while trying to increase individual or collective rewards - or Multi-Actor Learning - where multiple agents with the exact same goal collect experiences and share them for faster learning. But these

two scenarios correspond to the single-task setting. The former extends the notion of single-task to multiple agents, while the latter only uses multiple instances of the environment in order to parallelize training.

## B.3 Comparison

For differentiating between the three different paradigms of TL, MTL and Meta-Learning, it is important to understand that they are answers of increasing levels of abstraction (Upadhyay, 2023) to the question of how to exploit knowledge from previously learned tasks to solve new ones in only a few shots. TL, as the solution with the lowest level of abstraction, simply transfers the knowledge gained in one task (the source task) to a second one (the target task), while Meta-Learning, as the solution with the highest level of abstraction, generally tries to extract the core information to learn any task of a distribution $p$ in a few shots. In other words, Meta-Learning also extracts information from the source task(s) to use them as a prior for the target task(s), but this information can be much more general, and it results from the outer optimization rather than from solving the source task(s) in particular - a difference becoming even more unclear when foundation models get pre-trained on several source tasks in order to be meta-tested afterwards.

The most confusion in the literature is between MTL and meta-learning (Upadhyay, 2023) since both aim to solve several tasks as well as possible rather than exclusively focusing on one single target task (like in TL). While meta-learning iteratively and sequentially learns a number $N$ of tasks not necessarily fixed and, afterwards, tests the thus acquired knowledge on test tasks unseen during training, MTL trains a fixed number $N$ of tasks simultaneously and without testing the resulting model parameters $\theta$ on unseen tasks. The trained MTL model is, instead, evaluated on the validation set $X_{\text{test}}^i$ of each of the learned tasks, $T_1, \ldots, T_N$ which corresponds to the standard machine learning test phase but for $N$ tasks at the same time. In this way, MTL rather focuses on best solving the learned tasks than on adapting to new ones as fast and good as possible.

MTL can consist of tasks of different families, such as classification, regression, or segmentation, while meta-learning is normally reduced to one family of tasks represented by the distribution $p$ over tasks of the form (2). And, most importantly, MTL does not consist of an outer learner extracting prior knowledge (or meta representations). In this specific aspect, MTL is more similar to transfer learning, where prior knowledge between tasks is only transported implicitly via the model parameters. However, the TL paradigm particularly requires a target task to transfer the gained knowledge to, while the MTL paradigm does not specifically include such an option. This further highlights the similarity between TL and meta-learning as well as the difference between meta-learning and MTL.

# C Efficient Off-Policy Meta-RL

Efficient Off-Policy Meta-RL (PEARL) (Rakelly et al., 2019) is the landmark gradient-based task inference meta-RL algorithm and a frequently used benchmark for other meta-RL algorithms. In contrast to other extensions of MAML, such as $\alpha$-MAML (Behl et al., 2019) or PMAML (Finn et al., 2018), which only slightly modify the original paradigm, PEARL introduces a substantially different meta-learning scheme. It not only incorporates task inference (as also proposed in (Finn et al., 2018) or (Grant et al., 2018)), but also reformulates the meta-training scheme to off-policy learning. The main idea of PEARL is to infer a MDP-level latent context variable $z_i$ via Bayesian updates that encodes an MDP $M_i$ based on the current context (18). This context variable can either represent function approximations like the value function (6) (model-free approach), or the dynamics $\mathbb{T}_i$ and $R_i$ of the current MDP $M_i$ (model-based approach). It is sampled from a prior network $p_\varphi(z_i|c_i)$ at the beginning of each RL episode, whose weights $\varphi_z$ are learned at meta-level. During inner learning, the prior network updates implicitly through the context and thereby approximates the corresponding Bayesian posterior update (19).

The policy $\pi(\cdot|z_i)$ is a neural network with meta-learned weights $\varphi$. According to the BRL paradigm, it depends on the context variable $z_i$, i.e., on the belief about the current MDP. Hence, it implicitly adapts to a particular MDP through the context, so that a MDP-level gradient descent update of the policy weights $\varphi$ is not required. This makes PEARL off-policy: the policy $\pi_\varphi$ is only updated at meta-level, but remains unchanged throughout MDP-level learning. Instead, the belief updates based on the gathered experience.

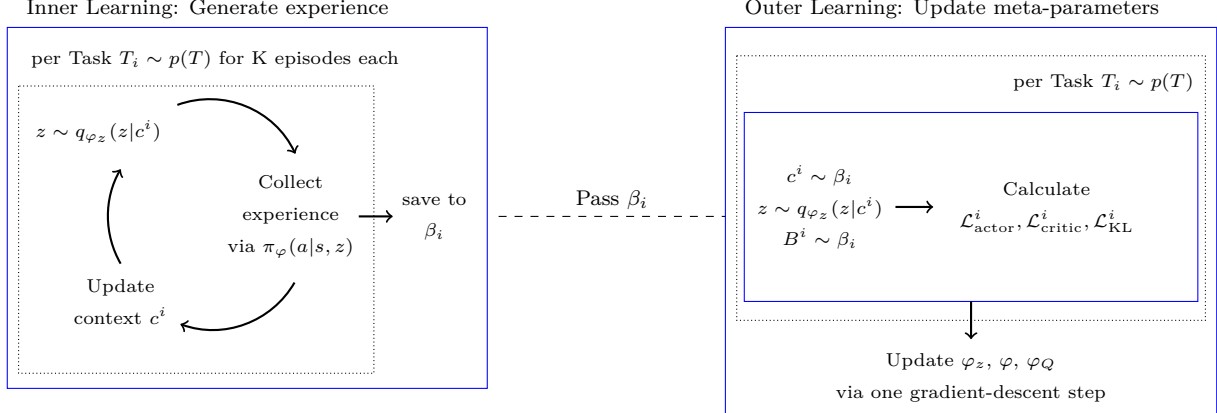

Figure 7: The PEARL meta-training scheme. In each MDP $M_I$, PEARL collects $K$ episodes of experience and stores them in replay buffer $\beta_i$. It samples the context variable $Z$ at the beginning of each episode. After inner learning, minibatches are sampled from the replay buffer to update meta-parameters $\varphi, \theta_\pi$ and $\theta_Q$.

In their original publication, (Rakelly et al., 2019) used Soft-Actor-Critic (Haarnoja et al., 2018). However, this is rather a design choice than a necessity of the PEARL paradigm, but it incorporates a critic $Q_{\varphi_Q}$ into the framework whose parameters $\varphi_Q$ must also be meta-learned via meta-gradient descent.

**Meta-Training and Meta-Testing**

Figure 7 illustrates the meta-training scheme of PEARL. At the beginning of each RL episode, $z_i$ is sampled from the prior distribution $q_{\varphi_z}(\cdot|c_i)$ conditioned on the current context $c^i$. Then, the policy $\pi_\varphi(\cdot|z_i)$ gathers one RL episode of experience $c_h^i$, and a MDP-specific replay buffer $\beta_i$ stores this experience for meta-level loss calculation. a MDP-specific test set $X_{\text{test}}^i$ is not required[18], As all meta-losses are calculated on the experience gathered and stored this way.

The meta-level updates of $\varphi$, $\varphi_z$ and $\varphi_Q$ are one meta-gradient descent step of the form (13), with the meta-losses for the actor $\pi_\varphi$ and critic $Q_{\varphi_Q}$ defined as the sum of the respective MDP-level losses. To stabilize training, the meta-loss for $\varphi_z$ contains a KL-divergence penalty

$$\mathcal{L}_{\text{KL}}^i := \mathrm{D}_{\text{KL}}(p(z_i \mid c_t^i), r(z_i))$$

with context $c_t^i$ sampled from replay buffer $\beta_i$. The baseline distribution $r$ is a hyperparameter of PEARL and chosen as a normal distribution in the original publication. The corresponding meta-loss for $\varphi_z$ is

$$\mathcal{L}_{\text{meta}}^{\varphi_z} := \sum_{M_i} \left( \mathcal{L}_{\text{critic}}^i + \mathcal{L}_{\text{KL}}^i \right).$$

It, additionally, contains the MDP-level critic losses, because the functions frequently used for critics (like (6)) highly depend on which MDP the agent is currently acting in.

The meta-testing scheme of PEARL is analogous to the general meta-testing scheme presented in Section 2.2. However, as the experience gathered during inner learning are saved in the replay buffer, no test episodes are required for performance evaluation. But, similar to meta-training, this is rather a design choice of (Rakelly et al., 2019) than a necessity.

**Performance Analysis**

PEARL outperforms MAML and RL$^2$ in terms of performance, adaptation speed and sample-efficiency on the MuJoCo benchmark (Todorov et al., 2012) - most likely due to its off-policy learning scheme. However,

---

[18]It is rather a design choice than a theoretical necessity to not sample another $K$ episodes with fixed $z_i$ or $p_{\varphi_z}$.

in contrast to more sophisticated meta-RL algorithms, such as VariBAD or TrMRL, it always requires two episodes of exploration (see, e.g., Zintgraf et al. (2020), Figure 6, or Melo (2022), Figure 5) before it performs comparatively well. This indicates a high adaptation speed and a high sample-efficiency, as well as poor zero-shot performance. The latter becomes even more severe in more complex tasks, such as Meta-World, where PEARL is not even able to solve tasks until the third episode ((Melo, 2022), Figures 4 and 5), which indicates poor exploration behaviour. This motivates more sophisticated gradient-based algorithms, such as DREAM (Liu et al., 2021), MAESN (Gupta et al., 2018) and MetaCure (Zhang et al., 2021), which are more tailored to optimally explore at the beginning of MDP-specific adaptation.

## D  Attention is all you need

The following paragraphs broadly discuss the architecture of the vanilla transformer (Vaswani et al., 2017). The goal is to give the reader an intuition how self-attention "creates context" and "focuses" on important information on the example of language-to-language translation. For a detailed description of the vanilla transformer architecture the interested reader is nevertheless referred to the original paper or (Alammar, 2018).

The vanilla transformer consists of an encoder and a decoder, which both divide into six 2- or 3-layer sub-blocks respectively[19]. Metaphorically speaking, the encoder translates an input sentence like

The child plays football because it likes it very much.

into "machine language", i.e. word embeddings, while the decoder translates these embeddings back into the desired output language. However, a word-by-word transformation from the input language into "'machine language "' and back into the output language preserves no information about what word the first "it" refers to and to which word the second "it" is related to. One needs a possibility to preserve the context between the words and, hence, the semantic structure of the sentence while encoding it. Similarly, while decoding, the syntactic structure of the sentence must be transformed so that it fits the grammar of the output language. For example, a German sentence has a totally different grammatical structure than its English counterpart with the same semantic meaning. In their original publication, (Vaswani et al., 2017) solved these problems using two techniques: Self-attention layers and positional encodings. The following paragraph discusses the former, while the latter is broadly discussed further below.

### D.1  Self-Attention

In a self-attention layer, every word embedding $x_i$ ($i = 1, 2, \ldots, n$) of a $n$-word input sentence $x$ like the one above gets a query, key and value projection matrix $Q_i, K_i$ and $V_i$ assigned to it. These matrices are parameters of the self-attention and, as such, get learned during training. They project a word embedding $x_i$ into query, key and value vectors $q_i, k_i$ and $v_i$ by multiplication. The "context" between words is then considered in the following way:

- Every word embedding $x_i$ gets "compared" to every other word embedding $x_j$ by multiplying every query vector $q_i$ with every key vector $k_j$ in a dot product. One can interpret the result as the context between the corresponding words of the input sentence since, mathematically, the dot product puts the query and key vectors in geometrical relationship to each other.

- For each word embedding $x_i$, the result of its query-key multiplication with every other word embedding (including itself) is a vector representing the attention scores of the word embedding $x_i$ to all other word embeddings $x_j$. This vector of attention scores is fed into a softmax function to yield the respective attention weights summing up to one.

- At last, these "attention weights" are multiplied by the value vector $v_i$, resulting in a weighted sum over attention weights with weights $v_i$ that directly depend on the learned projection $V_i$. Hence, the

---

[19]This number of six sub-blocks for encoder and decoder is rather an architectural choice by (Vaswani et al., 2017) than a necessity of the vanilla transformer.

model can learn which attention scores $q_i \cdot k_j$ are more or less important by adjusting the values of the value projection matrix $V_i$ respectively.

In practice, one typically simultaneously calculates all query-key pairs of the embedded input sentence $x$:

$$\text{Attention}(x) := \text{softmax}(\frac{QK^T}{\sqrt{d}}) \, V, \tag{21}$$

with the hyperparameter $d$ as the dimension of key, value and query vectors, $\sqrt{d}$ as the scaling factor and $K$, $V$ and $Q$ being the respective key, value and query matrices. The scaling factor $\sqrt{d}$ keeps the dot product logits small so that the soft-max does not slip into saturation. This maintains a proper gradient flow and thus stabilizes training.

Each encoder sub-block of the vanilla transformer consists of a self-attention layer so that the context between words (and hence the semantic structure of the sentence) is preserved throughout the whole embedding process. Encoding the above example sentence, the model can learn through the query-key multiplication $q_1 \cdot k_2$, that the first word "The" refers to the next word "child". At the same time, this connection is almost meaningless to the semantic of the whole sentence so that the model possibly learns to assign a very small value to the value vector $v_1$.

The self-attention layers of the decoder sub-blocks look slightly different to those described above. Since decoding the embeddings into output language is a sequential manner, no word can be set into context to words following later on in the sentence. Therefore, the corresponding query-key multiplications $q_i \cdot k_{i+a}$ with $a = 1, \ldots, n - i$ are set to minus infinity by construction so that the respective softmax results in zero attention weights.

In their original work, (Vaswani et al., 2017) implement multi-head-attention with the number of heads as a hyperparameter and show this to further boost performance. Multi-Head Attention duplicates the word embeddings $x_i$, projects them into the smaller subspace of each head [20], executes the Self-Attention of the form (21), , concatenates the outputs of all heads to one single output matrix, and projects them back into the original output dimension $d$ by another linear layer transformation.

## D.2 Positional Encoding

In the self-attention (21) the order of input tokens does not matter and hence the order of the words of the input sentence is not automatically preserved throughout encoding. However, the order of words in a sentence is quite important, especially as syntactic structure of input and output languages might differ, e.g. when translating English into German. This is why, (Vaswani et al., 2017) include a positional embedding for each word embedding $x_i$ in the vanilla transformer architecture that has the same length as the embedding itself. The form of the positional encoding $p_i$ is a hyperparameter of the vanilla transformer. In their original work, (Vaswani et al., 2017) combine different sin and cosine functions, but state that they have tried different positional encodings without a major loss in performance. Other transformer variants even learn this positional embedding function along with all the above model parameters.

---

[20] the queries, keys, values of each head are smaller-dimensional vectors.

