# OpenReview forum: "Meta-Learning and Meta-Reinforcement Learning - Tracing the Path towards DeepMind's Adaptive Agent"
_TMLR — Rejected by TMLR_

### Review · Reviewer_qvvT · 2025-12-29

**Summary Of Contributions:**

This paper surveys Meta-RL chronologically, trying to unify the field under a consistent task-based formalism. It traces evolution from gradient methods and recurrent approaches to Transformer-based models. The paper spends considerable effort connecting this history to DeepMind's Adaptive Agent (ADA), with discussion of curriculum learning and distillation.

**Audience:**

No

**Audience Explanation:**

This survey bridges classical Meta-RL literature and the emerging trend of generalist, transformer-based agents. The unified mathematical formalism provides a useful reference for researchers trying to understand theoretical connections between diverse algorithms (from MAML to foundation models). It should be valuable both as entry point for newcomers and as a synthesis for experts.

**Broader Impact Concerns:**

No concerns.

**Claims And Evidence:**

Yes

**Claims Explanation:**

For a survey paper, the evidence is mainly about accuracy of literature review and soundness of the mathematical framework. The Meta-RL formalism in Section 2 is derived correctly from standard supervised learning and POMDP definitions. The categorization of algorithms seems accurate and matches how the community understands these methods.

**Requested Changes:**

1. My main concern is the narrative feels too focused on building toward DeepMind's ADA. While ADA is impressive work, presenting the entire field as leading up to this one agent makes it read more like internal documentation than a neutral survey. I'd suggest broadenning Section 3.5 and Section 4 to treat other generalist agents (Gato, AMAGO, etc.) as peers rather than treating ADA as the culmination of everything.

2. There's a disconnect between the formalism you define and how you actually use it. Section 2.1 defines nice precise metrics like "Meta-Generalization Error" and "Adaptation Speed", but then when reviewing algorithms in Section 3, you mostly just give qualitative descriptions instead of analyzing them through these metrics. Would strengthen the paper if you tied discussion of MAML, VariBAD, etc. back to those mathematical objectives you defined.

3. Fix the spelling of "Deep Mind" to "DeepMind" in the title.

---

> ### Author Response · Authors · 2026-01-13
>
> We thank the reviewer for the highly valuable feedback.
> The Adaptive Agent is, indeed, only one representative of the family of generalist agents - although probably the most famous one. Hence, we agree that words like "`culminating"' are misleading when introducing ADA within the context of our timeline. We will adjust our formulations in Sections 1 and 3 accordingly to better reflect the representative role of ADA.
>
> Regarding your point, that the survey should be an entry point for newcomers and a synthesis for experts: This is what we were first aiming for. But the rebuttal phase of our first submission yielded a focus on an entry point for newcomers.
>
> Regarding the performance measures:
> We agree that there is a disconnect between our defined formalism and how we actually use it within our timeline. This is mainly due to a lack of comparability among literature (as we discuss in Section 4). Our main goal is to formalize what is implicitly used among different meta-learning publications, but never formalized. When reviewing the different landmark algorithms presented in our timeline, one needs to spend a considerable amount of time to fully understand the information presented within plots and figures - especially as a newcomer to the field.
> However, as we fully understand your criticism, we will go through the different landmarks again and determine their performances with the use of our defined measures as far as possible.
> Note, nevertheless, that benchmark task distributions increased in complexity along the evolution of meta-learning algorithms so that the comparability between algorithms probably remains an issue this work does not cover. We try to address this issue with Tables 1 and 2 and discuss it in Section 4.

---

### Review · Reviewer_aeWB · 2026-01-05

**Summary Of Contributions:**

This paper provides a task-based framing of meta-learning and meta-reinforcement learning that aligns notation and terminology across a range of existing methods. Rather than introducing new theoretical results, the paper employs lightweight mathematical abstractions to clarify commonly used concepts such as tasks, task distributions, inner and outer optimization, and evaluation procedures. Within this framing, it collects and standardizes performance notions such as adaptation speed, sample efficiency, and out-of-distribution evaluation. The main value of the work lies in its chronological organization of meta RL landmark algorithms under a unified conceptual lens, which helps contextualize the progression from early meta-learning methods to modern memory-based agents. Overall, the contribution is primarily pedagogical and its utility is strongest as a structured overview for readers entering the field.

Strengths:

- The attempt to formalize and standardize commonly used performance notions is valuable.

Weaknesses:

- The paper motivates its formalization by pointing to inconsistencies in evaluation practices. However, these claims are not substantiated with concrete examples. As a result, it remains unclear how severe these inconsistencies are in practice and to what extent the proposed formalization resolves them.
- While the paper employs mathematical notation, the formalism remains largely descriptive and is kept at a high level. Some assumptions are not clearly linked to the presented analysis (eg. why are fixed horizons needed?).
- Core concepts such as the inner/outer loop structure or meta-training vs. meta-testing are repeatedly restated across sections, leading to redundancy.

**Additional Comments:**

The organization of the text could be significantly improved with several concepts are repeated across sections and the exposition is often longer than necessary. I believe that some high-level ideas could be introduced once and referenced thereafter. Streamlining the presentation and reducing redundancy would substantially improve readability and lead to a better overall experience for the reader. Similarly, strengthening the case for the proposed evaluation measures would also be valuable.

**Audience:**

Yes

**Audience Explanation:**

The paper is likely to interest a subset of the TMLR readership that is new to the field or particularly value surveys. Though it is also important to note that the findings of this paper are quite limited and mostly focus on compiling information available in other sources.

**Claims And Evidence:**

Yes

**Claims Explanation:**

Mostly yes. The evidence in the paper is mostly indirect, relying on citations to prior work and narrative, with little explicit evidence in the form of comparative examples or demonstrations of failure modes under existing practices. Some claims are supported adequately, but several motivating claims are weakly supported or only implicitly justified.

Where the evidence is adequate:

- The proposed formalization does indeed work with the discussed algorithms.
- Claims about the structure and evolution of meta-learning and meta RL methods (eg. the progression from MAML to ADA) are accurate.
- High-level descriptions of how major algorithms operate (eg. gradient-based vs. memory-based) are consistent with the literature.
- Statements such as “method X tends to generalize better than method Y” are accompanied by citations and framed at a high level.

Where evidence is a bit weaker:

- The paper repeatedly asserts that evaluation practices in meta-learning and meta RL are inconsistent or informal, motivating the proposed formalization. However, no specific examples are shown. As a result, this motivation is asserted rather than demonstrated.
- Similarly, it is not demonstrated how the proposed evaluation protocols would improve reproducibility or lead to different conclusions than existing evaluation practices.

**Requested Changes:**

Notation:

- The statement that “the observation space $X$ and the horizon $h$ are generally assumed to be identical across tasks drawn from the same distribution..." is a bit unclear and potentially misleading. It is not obvious in what sense “identical” is meant (eg. this cannot refer to policy-induced state or observation distributions, which will generally differ across tasks).
- Moreover, the assumption of a fixed horizon across tasks does not hold in many RL settings, and I am not sure why it is even needed in this work. Why not drop it? Similarly, the comment "RL problems do not have to be episodic, but $h\leq \infty$ is a rather practical condition" is in my opinion unnecessary since a lot of simulated MDPs are infinite but just issue a truncation signal (which is a different thing than terminating early).
- In Eq. (3), “meta” is used as a superscript, whereas in the surrounding text it appears as a subscript.
- Eq. (6) assumes normalized loss functions, such that losses on unseen tasks are comparable in scale to those on meta-training tasks. While this assumption may hold for classification, it does not generally apply to regression or reinforcement learning objectives.
- Two different symbols (T) are used to denote transition functions in Section 2.1, which may cause confusion.
- The statement that MAML’s meta-training “typically achieves almost zero training loss and 100% training accuracy (indicating global convergence)” appears to be based on limited empirical evidence. To this end, if MAML indeed typically achieved perfect training accuracy, the motivation for more complex meta-learning methods is pretty unclear. Similarly, the claim that MAML cannot practically learn an infinite number of tasks due to finite network capacity applies to all meta-learning methods. While memory-based approaches may be more parameter-efficient, the currently presented argument risks overstating limitations as specific to MAML.

Presentation:

- Table 1 might be more effective if placed later in the manuscript, after the reader has been introduced to the relevant algorithms. Also, in its current form, the table contains a substantial amount of redundant information (eg. advantages such as “easy to implement,” “many open-source repositories available,” or “first-order implementations available”).
- Table 2 occupies considerable space while largely restating information already presented in Table 1, albeit organized by components rather than algorithms. The added value of this second table relative to Table 1 is therefore unclear.
- The manuscript uses `\citet` where `\citep` would seem more appropriate.

Redundancy:

- It is unclear why optimizers and initialization schemes are discussed in Section 3.1, as their relevance to the core meta-learning or meta RL concepts introduced there is not sufficiently motivated.
- Core concepts such as the general meta-learning/meta RL paradigm (inner vs. outer loop), the role of context in meta RL, and the exploration–exploitation trade-off are restated multiple times across sections, often with minimal variation. Similarly, the descriptions of meta-training versus meta-testing, limitations of RNN-based methods versus advantages of Transformers, and the motivation for uncertainty modeling recur across the RL2, VariBAD, and ADA sections. These concepts could be introduced once in a unified manner and subsequently referenced and contextualized in the individual algorithm descriptions.
- The paper contains some redundant mathematical formulations that could be streamlined without loss of clarity (eg. Eq. (1) and (2)).

Positioning:

- While the paper distinguishes meta-learning from closely related paradigms in Appendix B, I believe that a very short contrast of meta-learning with the neighboring ideas (eg. goal-conditioned RL, multi-task RL) would fit somewhere in the main body (however, this should come at the cost of deleting something else since the paper is already pretty long). For example, a short high-level summary, possibly framed in terms of which task components (eg. (T), (R), (X)) vary across tasks and how evaluation targets differ would improve conceptual clarity.
- The manuscript motivates its formal framework in part by claiming that performance measures in the meta-learning and meta RL literature are often treated informally and that concepts such as validation or meta-validation are rarely formalized. However, this claim is not supported by examples, references, or analysis demonstrating the prevalence or severity of these issues. Providing illustrative examples or citations would help clarify why this problem is significant and how the proposed formalization meaningfully addresses it.

---

> ### Author Response · Authors · 2026-01-13
>
> Thank you for your very thoughtful and valuable feedback. We are sure that it will help us a lot to improve the quality of our work. In the following, we directly respond to your bullet points:
>
> Notation:
> • Identical observation spaces"' often mean identical dimension/shape/format, dependent on the underlying task distribution. While RL tasks might require for the same information to be shared, classification probably relies on identical format of images among different tasks. We will add a small paragraph to sharpen our explanation in this respect. Note, however, that this distinction is a rather theoretical one, so that it highly depends on how one models the underlying meta-problem.
> Mentioning the horizon simply is for the sake of completeness. You are right that this is not relevant for application. We will delete/adjust accordingly. The same holds for the statement about episodic RL tasks.
> • We will, of course, adjust notation in Eq. (3). The same holds for notation of the transition.
> • Eq. (6): Thank you very much for this valuable comment. We fully agree and will sharpen our formulation. Of course, the losses must be within the same scale among tasks. In practice, this can easily be ensured by normalizing losses. For RL, normalized rewards and comparatively large horizons $h$ are required.
> •  The empirical findings about MAML's performance are taken from the work cited at the end of the sentence. But you are totally right that the sentence promises too much, and indeed, it seems to be a mistake that appeared during our last revision when shortening paragraphs. Of course, we talk about simple task distributions, here, and we will go through the cited reference in order to make our sentence more precise.
> Moreover, we agree on the misleading formulation regarding MAML's capability to scale to an infinite number of tasks. We mean that MAML is proven to be a universal approximator, but cannot be infinitely scaled. Thus, more parameter efficient methods are required, as you correctly stated. But for those methods, there not even is a universal approximation theorem. We will adjust our formulation accordingly in order to make this clearer.
>
> Presentation:
> • Tables 1 and 2 are, indeed, redundant. We will improve them along with getting rid of redundancies in Section 3.
> • Throughout the whole manuscript, we use \cite, so that the TMLR tex backbone defines the particular kind of bibtex style. To the best of our knowledge, this is how we are supposed to do it. But please further explain your comment, if we are mistaken.
>
> Redundancies:
> • Redundancies of Core concepts: As stated in our introduction: "`The paradigm of Section \ref{Sec_Paradigm} serves as the organizing thread: each new algorithm is situated within the same formalism"'. Hence, a little redundancy is on purpose for the sake of better structuring. We, nevertheless, fully understand the reviewer's concern - particularly regarding context, exploration/exploitation and other concepts not presented in Section 2 - and will thus try to further reduce redundancies.
> • Redundant mathematical formulations: The redundancy in Eq. (1) and (2) is on purpose. The idea throughout the whole paper is to always start with simple concepts that are generally known even for newcomers and modify/transfer them to more sophisticated, meta-learning specific notions. Here, (2) is the direct abstraction of (1). Please feel free to list other redundancies. We would be very happy to examine them, too.
>
> Conceptual scope:
> • Discussing initialization schemes and learning rate issues at the beginning of Section 3.1. aims to motivate gradient-based meta-learning. More precisely, placing initial parameters sufficiently close to a good task-specific solution for faster and more stable learning.
> • Related fields: In our first version submited to TMLR, the distinction between transfer-learning, Multi-Task-Learning and meta-learning was part of the main manuscript. But as all reviewers commented that this distinction is not required and distracts the reader from the main focus, we moved it into the Appendix.
> • performance measures: We agree that our performance measures are not well connected/utilized in Section 3, and we will go through the literature again to close this gap. Please also see our answer to the review below.

---

### Review · Reviewer_pQDF · 2026-01-06

**Summary Of Contributions:**

This work claims to unify the mathematical frameworks for various meta-learning approaches and trace the path from meta-learning to the ADA.

**Audience:**

No

**Audience Explanation:**

The paper is poorly written and littered with mathematical and grammatical errors. The authors did not put in the effort to improve the paper based on the thorough feedback from the previous round of reviews.

**Claims And Evidence:**

No

**Claims Explanation:**

The authors did not thoroughly address reviewer concerns from the previous reviewing round. The mathematical notation throughout the paper remains extremely poor and unclearly defined. Let's just take definition (1), as an example. The authors introduce (1) without immediately defining terms such as $\mu$ and the transition operator (these are deferred until Example 1). $X_{train}$ is asserted as existing but doesn't appear in the task definition $T$ -- this doesn't make sense as $\mu$ depends on $X_{train}$! In Example 1, the authors write "Here, [the transition function] is a unit distribution conditioned on all images selected so far since all remaining images in $X_{train}$ are always equally likely to be chosen next." But they define $T: \mathcal{X} \times \mathcal{X} \to \mathbb{R}$ -- there's no notion of "distribution" here at all! Presumably, the authors mean that $T$ maps from $\mathcal{X}$ to a discrete probability distribution over $X_{train}$. Not to mention that the sentence they wrote is quite difficult to interpret, mathematics notwithstanding. I suppose they mean that, conditioned on images already produced, one of the remaining images in $X_{train}$ is selected uniformly at random. Of course, this is not possible to express with just a function $T: \mathcal{X} \times \mathcal{X} \to \mathbb{R}$.

The remainder of the paper has borderline unintelligible mathematics along these lines. It needs to be completely and carefully reworked.

**Requested Changes:**

The authors should properly address the last round of reviews (especially writing quality concerns) instead of spamming resubmissions.

---

> ### Author Response · Authors · 2026-01-21
>
> For classification, this distinction is rather technical: In the case, where one samples with replacement, subsequent samples are independent of the current one. In the case of sampling without replacement, defining a conditioned probability would be rather complicated, as the reviewer correctly stated. However, the second definition is much simpler, since it considers observation pairs instead of current and subsequent observation. In this case, we can define $\mathbb{T}: \sigma((\mathcal{X} \times \mathcal{X})\setminus D) \to [0,1]$, where $D:=\{ (x,y) \in \mathcal{X} \times \mathcal{X}: x=y \}$ is the set of diagonal pairs, i.e. means sampling the same image twice.
> Of course, one must also exclude images that were sampled before the current observation pair, but this can simply be done by defining $D$ in a more general way. One obtains a special multivariate hypergeometric distribution, where each image corresponds to a different event (i.e. the population has only individuals, no groups). And for a very large observation space, this is even multinomial.
>
>
>
> We thank the reviewer for the feedback on our manuscript.
>
> We agree that Definition (1) and Example 1 were unclear in the previous version, in particular regarding the formal role of the transition component. Specifically, the presentation did not sufficiently distinguish between observation sampling in supervised learning and state transitions in sequential decision-making settings, which led to ambiguity in the interpretation of the notation.
>
> To address these issues, we have revised Section 2 as follows:
>
> - Definition (1) has been revised so that all components are introduced and defined explicitly at first occurrence. In particular, we clarify the intended interpretation of the transition component and its relation to observation sampling, avoiding the previous conflation of functions and probabilistic notions.
> - In Example 1, we now explicitly state that, in the supervised-learning setting, the transition, initial distribution and horizon do not play a meaningful role. In the previous version, they were mainly added for the sake of completeness. However, considering the reviewer's justified criticism, we admit that this distracts the focus from the core concepts.
> - We have revised the surrounding text to define all task components in a general and consistent manner, improving clarity and avoiding deferred or implicit definitions.
>
>
> Finally, we have strengthened the connection between the formal framework in Section 2 and the historical timeline in Section 3, clarifying how the proposed abstraction relates to existing meta-learning and meta-RL approaches.

---

### Decision · Action_Editor_Mqnk · 2026-02-11

**Recommendation:** Reject

**Audience:**

Yes

**Audience Explanation:**

The topic of the paper is timely and interesting and the aim of the paper, if executed well, would add a very valuable contribution, particularly for newcomers diving into the complex set of methodologies developed in meta-learning and meta-RL, and how they are connected to modern deep learning practice. As such there is a potentially large audience for the paper. I do partly agree with aeWB (in their final recommendation): "While there is some niche audience for the manuscript in the current form, the audience could be substantially broadened by focusing the story more".

**Claims And Evidence:**

No

**Claims Explanation:**

The submitted article aims at presenting a comprehensive review of algorithmic and methodological developments in meta-learning and meta-RL with the goal of helping newcomers to the field to understand and trace how the distinct developments led to the various components of DeepMind's adaptive agent or how they can be seen as alternatives (or relevant historic developments). This is a resubmission which was originally criticised for lacking rigor, clarity, and focus.

While I do consider the current manuscript a step in the right direction, large parts of the previous submission remain unmodified, and the overall length of the manuscript went from 29 to 25 pages. The suggestion of a *major revision* in the decision of the previous manuscript was only partly and superficially fulfilled. The current manuscript still lacks (like the previous manuscript) in terms of rigor and focus: the goal to provide a gentle introduction to understanding the adaptive agent, is complicated by adding many irrelevant parts of the historic developments and alternatives, all of which can be trimmed from the main body into the appendix---newcomers do not require dozens of pages to understand the core methodology and components of the adaptive agent. Additionally, the ambition of providing a unified mathematical treatment in section 2 is good in principle, but currently lacks rigor and clarity. It is hand-wavy (see pQDF's criticism for one example, which was partly addressed to be fair) and uses non-standard terminology and nomenclature throughout, which makes it harder instead of easier for newcomers. Most of the material in Section 2 is textbook material (or has been described in excellent reviews or individual papers)---I strongly suggest the authors follow a ML and RL textbook very closely for these parts. Just to name one example of many, the notion of 'standard ML' is itself non-standard; it would be more clear to start with introducing the (standard) framework of empirical risk minimization (where the notion of generalization error on P.10 comes from and is formally justified) and then extend it to the meta-case.

The current manuscript has had a fresh set of 3 reviewers compared to the previous submission, and after the rebuttal and author discussion, two of them are suggesting a 'reject' and one is 'leaning accept'. One reviewer criticised that the authors have not sufficiently addressed the previous rounds of reviews before resubmitting, particularly in terms of mathematical rigor and clarity. I do think the resubmission has improved, but I also agree to some extent with that reviewer (and the excessive length of the work, which causes significant workload on the reviewers' end adds additionally to the issue of a hasty resubmission). Another reviewer suggests in their final recommendation that the authors should address the criticism raised in all reviews, and that they believe that this would significantly broaden the target audience of the paper. The third reviewer is happy with the changes made during the rebuttal, but is only weakly in favor of accepting the paper. Taking all of this together, I do not think that the paper has sufficiently improved compared to the previous submission to pass the bar for acceptance. I stand by my original assessment that a lot of interesting work has gone into the manuscript, and that I believe it is worth spending the additional time and effort to polish this into an impactful piece for newcomers to (memory-based) meta-RL. One option could be to focus entirely on explaining memory-based meta-RL, relate it to the adaptive agent, and reduce the remaining discussion of meta-learning in general and gradient-based methods to two or three paragraphs (the current text can be added as an appendix). Additionally, the current section 2 can be trimmed (e.g., are the examples needed in full detail?) and it needs another careful pass for clarity, correctness, and alignment with standard terminology/frameworks.